# Towards Accurate Time Series Forecasting via Implicit Decoding

**Xinyu Li[1], Yuchen Luo[1], Hao Wang[2], Haoxuan Li[3,6],**
**Liuhua Peng[1], Feng Liu[1], Yandong Guo[4], Kun Zhang[5,6], Mingming Gong[1,6]***
[1]The University of Melbourne  [2]Zhejiang University  [3]Peking University  [4]AI[2] Robotics
[5]Carnegie Mellon University  [6]Mohamed bin Zayed University of Artificial Intelligence
xl5@student.unimelb.edu.au,mingming.gong@unimelb.edu.au

## Abstract

Recent booming time series models have demonstrated remarkable forecasting performance. However, these methods often place greater focus on more effectively modelling the historical series, largely neglecting the forecasting phase, which generates long-term forecasts by separately predicting multiple time points. Given that real-world time series typically consist of various long short-term dynamics, independent predictions over individual time points may fail to express complex underlying patterns and can lead to a lack of global views. To address these issues, this work explores new perspectives from the forecasting phase and proposes a novel Implicit Forecaster (IF) as an additional decoding module. Inspired by decomposition forecasting, IF adopts a more nuanced approach by implicitly predicting constituent waves represented by their frequency, amplitude, and phase, thereby accurately forming the time series. Extensive experimental results from multiple real-world datasets show that IF can consistently boost mainstream time series models, achieving state-of-the-art forecasting performance. Code is available at this repository: `https://github.com/rakuyorain/Implicit-Forecaster`.

## 1 Introduction

Time Series Forecasting (TSF) is the task of predicting future trends and dynamics of time series based on historical observations. It is essential for many real-world applications, including weather forecasting [52, 28], energy management [21, 3], and traffic flow estimation [1, 32]. In the past few years, the advancement of deep learning has enabled neural forecasting methods to take a significant place in the TSF landscape. Neural networks, especially Recurrent Neural Networks (RNNs) [10, 11], are particularly effective on TSF due to their ability to capture sequential information. Convolutional Neural Networks (CNNs) [12, 51], with their strength in recognising intricate temporal structures and local patterns, excel at extracting features from time series. By modelling dependencies between a sequence of representations, Transformers have demonstrated superior ability in learning inter-series [13] and intra-series [20] information, thus becoming state-of-the-art in TSF.

Despite these achievements, existing TSF methods primarily focus on improving the modelling of historical time series, while more precisely translating the modelled features into future series during the forecasting phase is rarely considered. Specifically, previous works typically use elaborately designed architectures to learn and transform the input time series into complex representations, and then leverage MLPs to predict these representations separately into the value of each future time step, thereby generating the output series. We noticed that this scheme of independent prediction between consecutive points might not be suitable for long-term forecasting, given that native MLPs lack a

---

*Corresponding author

39th Conference on Neural Information Processing Systems (NeurIPS 2025).

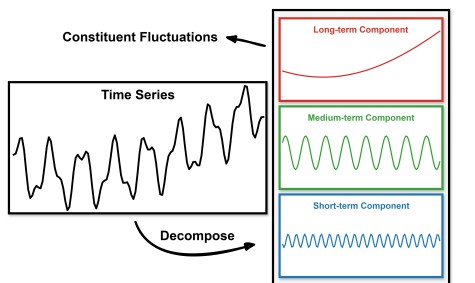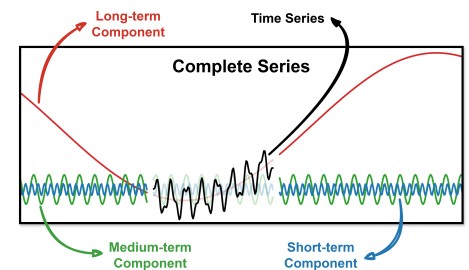

Figure 1: Decomposition of a time series into sub-components. *Left:* illustrates the breakdown of a simple time series into its constituent parts, each reflecting fluctuations over different frequencies. *Right:* provides a complete view of each individual component to clearly visualise how the combination of different periodic fluctuations forms the observed time series.

global view of time series' overall fluctuations [45] and are unable to handle the autocorrelation among predicted points, which may hinder them from forecasting and expressing general trends and long-term patterns. Besides, real-world time series often exhibit a mixture of long short-term dynamics, yet point-wise MLPs directly predict the intertwined original series while ignoring this compositional structure. This results in a forecasting process that lacks the understanding of underlying patterns, which not only increases forecasting difficulty but can also produce indistinguishable and uninterpretable predictions over various dynamics.

Considering the numerous underlying dynamics, it is apparent from Figure 1 that a time series can be regarded as a combination of multiple different types of fluctuations, where long-period fluctuations can reflect the overall trend of the time series and short-period fluctuations can indicate the seasonality. Extensive research [41, 54, 49, 36] has proven that forecasting future time series by separately predicting its trend and seasonal components can enhance performance. This decomposition forecasting approach further highlights the potential to forecast future series by predicting more nuanced trends and dynamics. Consequently, we pose the following question: ***Is it feasible to form the future time series by globally predicting constituent fluctuations across different periodicities?***

In light of the aforementioned problems and motivations, we explored the forecasting phase of TSF, focusing on accurately translating the learned information into actual forecasts. In this paper, we propose Implicit Forecaster (IF), a novel decoding module that replaces the output layers of previous TSF methods. IF offers a different forecasting perspective: instead of predicting future time points separately, it globally forms the time series by combining various base fluctuations, where these fluctuations are treated as periodic waves that can be implicitly represented and predicted just by their frequency, amplitude, and phase. Experimentally, IF is compatible with recent mainstream TSF models and can consistently boost their forecasting performance, making them state-of-the-art on 14 real-world datasets widely used in TSF and improving the result by a large margin on several of them. The contributions of our work lie in three folds:

- We refine the forecasting phase of TSF and propose *Implicit Forecaster* to replace the output layers of existing TSF models. It globally predicts various constituent waves from a frequency perspective to achieve accurate long-term time series forecasting.

- Experimentally, the proposed *Implicit Forecaster* can be applied to mainstream TSF models, boosting their performance by a large margin and achieving consistent state-of-the-art on multiple real-world benchmarks.

- We show that a better forecasting phase also matters for accurate TSF, an aspect largely overlooked by existing research. We hope our work can draw research attention to this critical point and inspire novel advancements.

## 2 Related Work

### 2.1 Time Series Forecasting with Deep Models

In the rapidly evolving field of time series forecasting, deep learning-based methods have garnered significant attention and attained remarkable success. RNNs have been favoured for their proficiency in handling sequential data. LSTNet [10] integrates convolutional layers into recurrent structures, effectively learning long short-term patterns. SegRNN [11] reduces recurrent steps by leveraging Segment-wise Iterations and Parallel Multi-step Forecasting strategies, achieving notable performance and efficiency. CNNs excel in learning local temporal patterns underlaid in time series, where MICN [34] employs multiple convolutional kernels to capture local features and global correlations, offering holistic views of time series. TimesNet [40] extends 1D time series into 2D space for better time series representation, thereby extracting multi-periodicity information. MLPs, built upon linear transformations, have demonstrated impressive forecasting accuracy while maintaining high computational efficiency. TimeMixer [36] utilizes a decomposable multiscale mixing, capturing temporal patterns at different granularities. SOFTS [4] introduces the STar Aggregate-Redistribute (STAR) module, aggregating global information across channels to capture inter-series correlations.

Transformer [27], as the most prominent sequence modelling architecture that achieved exceptional success in natural language processing [39] and computer vision [2], has emerged as the most popular approach for solving TSF problems. Autoformer [41] combines traditional time series decomposition with auto-correlation to extract better predictive components. PatchTST [20] adopts a Vision Transformer-like strategy, segmenting time series into smaller patches to better model intra-series dependencies. Crossformer [50] and CARD [37] explore interdependencies between patches across channels, concurrently learning cross-time and cross-variate information. iTransformer [13] treats the entire time series as an individual token to model series-wise dependencies, thereby explicitly learning multivariate correlations.

Given that time series forecasting is an end-to-end process, previous methods predominantly concentrated on optimizing the "historical series to representation" stage. Our work, however, shifts the focus to the "representation to predicted series" stage, bridging the gap in the forecasting phase.

### 2.2 Time Series Forecasting with Fourier Analysis

Fourier analysis offers methods for effectively modelling the underlying dynamics within time series. Recent advancements demonstrated that integrating frequency-domain techniques can enhance the performance of TSF. FEDformer [54] employs frequency-enhanced attention for efficient computation while better capturing the global properties. FiLM [55] applies Legendre Polynomial projections to approximate historical information and remove noise. FreTS [45] applies MLPs directly to the frequency domain, learning the mappings of time series in the complex plane. FreDF [30] proposes a new loss term to penalize forecasting errors within the frequency domain, enhancing the autocorrelation between predicted points. OLinear [46] operates in an orthogonally transformed domain to better model time series and utilizes a normalized weight matrix to efficiently capture multivariate correlations. FreEformer [47] further leverages a frequency-enhanced Transformer to model cross-variate dependencies in the complex domain, introducing an enhanced attention mechanism that improves feature diversity and gradient flow. In contrast to these methods, which primarily focus on leveraging Fourier analysis to model the historical time series, our approach explores a novel strategy that incorporates frequency techniques in the forecasting phase, aiming to achieve more accurate pattern predictions and address the lack of global views of output series.

## 3 Methodology

**Problem Definition** Consider a $T$ length multivariate time series with $N$ number of variates, represented as $\mathbf{X} = \{\mathbf{x}_1, \mathbf{x}_2, \ldots, \mathbf{x}_T\} \in \mathbb{R}^{T \times N}$, where each $\mathbf{x}_t \in \mathbb{R}^N$ corresponds to $N$ concurrent variates observed at time $t$. Also, consider the subsequent time series of $\mathbf{X}$ over a future horizon $L$, denoted as $\mathbf{Y} = \{\mathbf{x}_{T+1}, \mathbf{x}_{T+2}, \ldots, \mathbf{x}_{T+L}\} \in \mathbb{R}^{L \times N}$. Given a forecasting model $f_\theta$ parameterized by $\theta$ that maps the historical time series $\mathbf{X}$ to its future forecast as $\hat{\mathbf{Y}} = f_\theta(\mathbf{X})$, where $\hat{\mathbf{Y}} = \{\hat{\mathbf{x}}_{T+1}, \hat{\mathbf{x}}_{T+2}, \ldots, \hat{\mathbf{x}}_{T+L}\} \in \mathbb{R}^{L \times N}$, the objective of time series forecasting is to optimize $\theta$ such that the forecast $\hat{\mathbf{Y}}$ closely approximates the true future $\mathbf{Y}$.

## 3.1 Implicit Forecaster

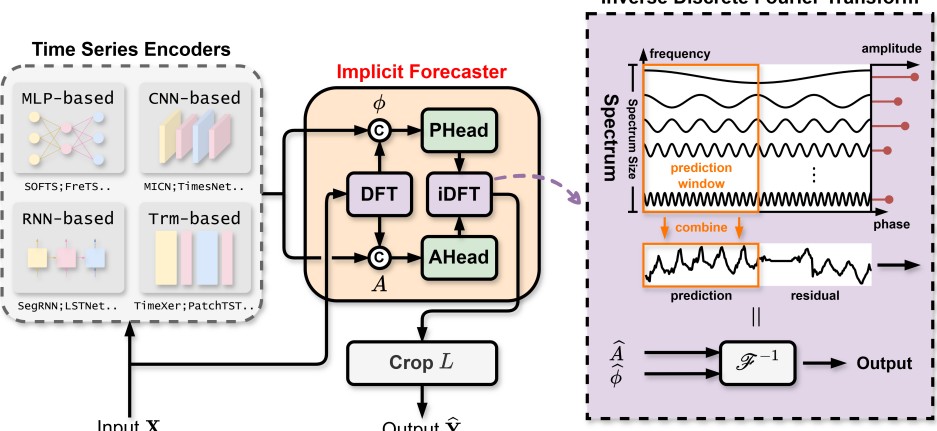

Figure 2: The overall architecture of Implicit Forecaster. The input time series will first be learned and converted into hidden representations via the encoder part of other time series models. The proposed Implicit Forecaster then decodes such representations into amplitudes and phases of various periodic waves, composing them into output series.

As depicted in Figure 2, our Implicit Forecaster mainly consists of an Amplitude Head (AHead), a Phase Head (PHead), a Discrete Fourier Transform (DFT) module, and an inverse DFT (iDFT) module. We will dissect its architecture around how it generates the forecasts in this section.

**Encoder Integration**  IF receives two inputs for forecasting: 1) a hidden encoder representation that can maximally capture the inherent information and temporal features of the input time series, and 2) the original input time series provided by a skip-connection. Since IF mainly serves as an additional decoding module for improving the forecasting phase, we take existing forecasting models (e.g., the encoder part of TSF Transformers) as the time series encoder for learning the encoder representation, and replace their output layers with IF. Given a multivariate time series $\mathbf{X} \in \mathbb{R}^{T \times N}$, learning the encoder representation can be easily formulated as follows:

$$\mathbf{X}_{enc} = \text{TimeSeriesEncoder}(\mathbf{X}), \tag{1}$$

where $\text{TimeSeriesEncoder} : \mathbb{R}^{T \times N} \mapsto \mathbb{R}^{N \times D}$ is the encoder module, $\mathbf{X}_{enc} \in \mathbb{R}^{N \times D}$ is the encoder representation of the input time series and $D$ stands for the model dimension. Note that the encoder representation is channel-separated over $N$, as IF independently forecasts each variate of time series.

**Implicit Decoding**  In the forecasting phase, rather than directly predicting the learned information into each output time point like conventional methods, IF implicitly predicts the constituent fluctuations of the future time series. Concretely, for a desired forecast, IF presupposes a **frequency pool** (i.e., a fixed-size spectrum) containing waves with different long short-term frequencies and separately predicts the amplitude and phase of each wave. These implicitly represented waves will serve as the constituent components and will be combined as the future time series, where the part within the forecasting length will be cropped as the final output. Additionally, IF uses a skip-connection to extract and incorporate the amplitude and phase information of the original input series, given that such information is likely to persist throughout future observations, which will be concatenated to the encoder representation as a supplementary feature map to assist the wave prediction.

Taking the encoder representation $\mathbf{X}_{enc} \in \mathbb{R}^{N \times D}$ and the historical time series $\mathbf{X} \in \mathbb{R}^{T \times N}$ as its input, IF forecasts the future time series $\hat{\mathbf{Y}} \in \mathbb{R}^{L \times N}$ as follows:

$$\begin{aligned}
\mathbf{S} &= \mathcal{F}(\mathbf{X}^{\top}), \\
\hat{\mathbf{A}} &= \text{AHead}(\text{Concat}(\mathbf{X}_{enc}, |\mathbf{S}|)), \\
\hat{\boldsymbol{\phi}} &= \text{PHead}(\text{Concat}(\mathbf{X}_{enc}, \arg(\mathbf{S}))), \\
\hat{\mathbf{Y}} &= \text{Crop}_{:L}(\mathcal{F}^{-1}(\hat{\mathbf{A}} \cdot e^{j\hat{\boldsymbol{\phi}}})^{\top}),
\end{aligned} \tag{2}$$

where $\mathcal{F}$ denotes the Discrete Fourier Transform and $\mathcal{F}^{-1}$ is its inverse operation, and $j$ is the imaginary unit defined as the square root of $-1$. $\mathbf{S} \in \mathbb{C}^{N \times T}$ is a complex matrix denoting the spectrum of the original input $\mathbf{X}$. $|\mathbf{S}|$ is the amplitude of each constituent wave from $\mathbf{X}$ and arg is the argument operation that computes the phase of each constituent wave. AHead is the amplitude predictor and PHead is the phase predictor. Suppose the frequency pool of IF contains $P$ waves with different frequencies to form the future time series, then $\hat{\mathbf{A}} \in \mathbb{R}_{\geq 0}^{N \times P}$ and $\hat{\boldsymbol{\phi}} \in [-\pi, \pi]^{N \times P}$ denote the predicted amplitudes and phases of these waves, respectively. We provide a detailed discussion of IF in the next section.

## 3.2 Overall Workflow

In this section, we provide a comprehensive introduction to the internal components of Implicit Forecaster, explaining the motivations behind various designs.

**Discrete Fourier Transform and its Inverse**    IF aims at forming time series with various periodic waves, while the DFT [26] and its inverse provide effective tools for representing and processing these waves. By transforming a signal from the time domain into the frequency domain, DFT enables the decomposition of a time series into its constituent frequency components, each represented by a complex number that encodes both the amplitude and phase of the corresponding sinusoidal wave at a certain frequency. Consider a univariate time series $\mathbf{x} = \{x_1, x_2, \ldots, x_N\} \in \mathbb{R}^N$, we can convert it to its spectral representation $\mathbf{z} = \{z_1, z_2, \ldots, z_N\} \in \mathbb{C}^N$ with $N$ waves by DFT as follows:

$$z_k = \sum_{n=1}^{N} x_n \cdot e^{-j2\pi k \frac{n-1}{N}} = |z_k| \cdot e^{j \arg(z_k)} = A_k \cdot e^{j\phi_k}, k = 1, \ldots, N, \qquad (3)$$

where $z_k \in \mathbb{C}$ corresponds to the complex of $k$-th wave components with a frequency of $\frac{k-1}{N}$. By considering each wave component $z_k$ in a polar coordinate, $|z_k| = A_k = \sqrt{\mathrm{Re}(z_k)^2 + \mathrm{Im}(z_k)^2} \in \mathbb{R}_{\geq 0}$ represents its amplitude and $\arg(z_k) = \phi_k = \mathrm{atan2}(\mathrm{Im}(z_k), \mathrm{Re}(z_k)) \in [-\pi, \pi]$ represents its phase, where $\mathrm{Re}(z_k), \mathrm{Im}(z_k) \in \mathbb{R}$ are the real part and the imaginary part of $z_k$, respectively.

On the other hand, the inverse DFT allows for the reconstruction of a time series from provided frequency components with known amplitudes and phases. This feature is particularly critical for our approach, as IF essentially seeks to predict amplitudes and phases of various waves with different frequencies to implicitly represent underlying fluctuations of future time series, where the inverse DFT is used to combine these waves to obtain the final forecast. Reversely, given amplitudes $A = \{A_1, A_2, \ldots, A_N\} \in \mathbb{R}_{\geq 0}^N$ and phases $\phi = \{\phi_1, \phi_2, \ldots, \phi_N\} \in [-\pi, \pi]^N$ of $N$ waves, we can construct a time series signal $\mathbf{x} = \{x_1, x_2, \ldots, x_N\} \in \mathbb{R}^N$ through inverse DFT as follows:

$$x_n = \frac{1}{N} \sum_{k=1}^{N} z_k \cdot e^{j2\pi n \frac{k-1}{N}} = \frac{1}{N} \sum_{k=1}^{N} A_k \cdot e^{j\phi_k} \cdot e^{j2\pi n \frac{k-1}{N}}, n = 1, \ldots, N, \qquad (4)$$

where $x_n \in \mathbb{R}$ is the $n$-th value in $\mathbf{x}$ and $z_k \in \mathbb{C}$ is the complex of the $k$-th wave.

**Frequency Pool**    Based on DFT properties, any length-$L$ time series can be completely represented as a linear combination of $L$ wave components, where each wave corresponds to a normalized frequency of $f_k = k/L$, for $k = 0, \ldots, L-1$. Therefore, it is possible for IF to forecast such time series just by predicting the amplitudes and phases of these $L$ waves, while we can define $\mathcal{P}_f = \{f_0, f_1, \ldots, f_{L-1}\}$ as the frequency pool of IF. However, we observed that there may exist very long-period fluctuations in time series, such that the input time series or the forecast can only encompass a small portion of their complete cycle. Given that these fluctuations operate at lower frequencies than those that can be directly predicted by IF (i.e., beyond the IF's frequency pool), they can only be approximated and represented by blending the pool's existing frequencies (a phenomenon known as *Spectral Leakage*), which complicates the forecasting process.

To better capture these ultra-low-frequency components, we extend IF's frequency pool. Specifically, when forecasting a time series of length $L$, we consider increasing the size of the frequency pool to $P$ ($P \geq L$), such that it contains frequencies $f_k = k/P$ for $k = 0, \ldots, P-1$. Consequently, IF predicts the amplitudes and phases of $P$ waves rather than $L$, allowing it to directly access and predict lower-frequency components. This extension enhances the representational capacity of IF, enabling

more accurate forecasting of time series with large-scale or slow fluctuations. Moreover, according to the conjugate symmetry property of the DFT, the spectrum of a real-valued time series is symmetric, where the second half is the complex conjugate of the first half. Therefore, in practical implementation, IF only needs to predict the first half of its frequency-domain components to reconstruct the complete time signal, substantially reducing computational cost.

**AHead**  The Amplitude Head is essentially an MLP that maps the encoder representations to the amplitudes of the waves in the frequency pool. To enhance information utilization, it also incorporates the amplitude information derived from the original input time series, thereby obtaining a global view of the input's energy distribution and improving the accuracy of future amplitude prediction. Given the $n$-th variate's encoder representation $\mathbf{X}_{enc,n} \in \mathbb{R}^D$ and corresponding input amplitude information $\mathbf{A}_n \in \mathbb{R}^T_{\geq 0}$, the AHead can be formulated as follows:

$$\text{AHead}(\mathbf{X}_{enc,n}, \mathbf{A}_n) = |\text{MLP}_{amp}(\text{Concat}(\mathbf{X}_{enc,n}, \mathbf{A}_n))|, \tag{5}$$

where $\text{MLP}_{amp} : \mathbb{R}^{D+T} \mapsto \mathbb{R}^P$ is a dense network with two layers for predicting the amplitude.

**PHead**  The Phase Head is responsible for predicting the relative position of the waves in the frequency pool, ensuring they are properly aligned within the forecasting window. However, directly regressing raw phase values is challenging due to the inherent numerical discontinuity of phases at their boundaries, where abrupt changes may occur from $\pi$ to $-\pi$. Such discontinuities make it difficult for IF to produce stable and consistent phase predictions. To mitigate this issue, we adopt a continuous representation of phases by predicting their sine and cosine components instead. Predicting these components ensures phases have a smooth transition across their periodic boundaries, thereby facilitating a more accurate phase prediction. Therefore, taking the encoder representation $\mathbf{X}_{enc,n} \in \mathbb{R}^D$ and input phase $\boldsymbol{\phi}_n \in [-\pi, \pi]^T$ of the $n$-th variate as its input, the PHead consists of two MLPs and can be formulated as follows:

$$\hat{\boldsymbol{\alpha}}_n = \text{Tanh}(\text{MLP}_{sin}(\text{Concat}(\mathbf{X}_{enc,n}, \boldsymbol{\phi}_n))),$$
$$\hat{\boldsymbol{\beta}}_n = \text{Tanh}(\text{MLP}_{cos}(\text{Concat}(\mathbf{X}_{enc,n}, \boldsymbol{\phi}_n))), \tag{6}$$
$$\hat{\boldsymbol{\phi}}_n = \text{atan2}(\hat{\boldsymbol{\alpha}}_n, \hat{\boldsymbol{\beta}}_n),$$

where $\hat{\boldsymbol{\phi}}_n \in [-\pi, \pi]^P$ is the predicted phases of pool waves. $\hat{\boldsymbol{\alpha}}_n, \hat{\boldsymbol{\beta}}_n \in [-1, 1]^P$ are the sine and cosine components, and $\text{MLP}_{sin}, \text{MLP}_{cos} : \mathbb{R}^{D+T} \mapsto \mathbb{R}^P$ are their corresponding predictors.

## 4  Experiments

We conduct extensive experiments to evaluate the performance of mainstream TSF models equipped with IF across various datasets and forecasting scenarios, comparing them with their original results to analyze the effectiveness of IF.

**Datasets**  We comprehensively include 14 benchmark datasets commonly used in TSF for our experiments, covering various real-life domains such as energy, traffic, weather, economics, and disease. Specifically, these datasets are ETT (Electricity Transformer Temperature) with 4 subsets, ECL (Electricity Consuming Load), Traffic, Weather [53], Exchange Rate, ILI (Influenza-Like Illness) [41], PeMS (Traffic data of Caltrans Performance Measurement System) with 4 subsets [12], and Solar Energy [10]. We split each dataset into training, validation, and test sets in respective ratios of 70%, 15%, and 15%, with all datasets divided strictly in chronological order to prevent data leakage issues. The validation set is used for monitoring the training loss of models, and the test set evaluates the model's final performance quantitatively. Lastly, all datasets are preprocessed with standardisation based on the training set. Further descriptions for each dataset are provided in Appendix A.1.

**Baselines**  We carefully select 7 well-acknowledged TSF models as our benchmark methods for comparison. These methods are from a diverse range of model types (we choose the most competitive method of each type), including: 1) one MLP-based method: SOFTS [4], 2) four Transformer-based methods: TimeXer [38], iTransformer [13], PatchTST [20] and Crossformer [50], 3) one RNN-based method: SegRNN [11], and 4) one CNN-based method: TimesNet [40]. These models will be paired with IF to compare against their original performance, with all models being evaluated under the same framework and environment. We provide a detailed introduction to baselines in Appendix A.2.

**Implementations**   All models are implemented entirely in Python and built upon PyTorch 2.0, with baseline methods directly adopted from their official implementations. The experiments reported in this paper were conducted on a 16-core AMD EPYC 9654 CPU and a single NVIDIA RTX 4090 GPU. We choose Adam optimizer [9] and L2 loss to learn the model parameters and take MSE (Mean Squared Error) and MAE (Mean Absolute Error) as metrics to evaluate the models. More implementation and experimental details are provided in Appendix A.3.

## 4.1   Main Results

Table 1 presents the forecasting performance of all models across 14 benchmarks, including their results after being equipped with the proposed Implicit Forecaster (**w/ IF**), where better values are **bolded**. Since MSE/MAE measures the discrepancy between the forecast and the ground truth, a lower value indicates better model performance.

Table 1: Multivariate time series forecasting results, averaged over respective prediction lengths $L \in \{24, 36, 48, 60\}$ for the ILI dataset, $L \in \{12, 24, 48, 96\}$ for the PEMS datasets, and $L \in \{96, 192, 336, 720\}$ for the remaining datasets. The lookback window length is set to $T = 36$ for the ILI and $T = 96$ for the others.

| Models | SOFTS (2024) | | w/ IF (Ours) | | TimeXer (2024) | | w/ IF (Ours) | | iTransformer (2024) | | w/ IF (Ours) | | SegRNN (2023) | | w/ IF (Ours) | | PatchTST (2023) | | w/ IF (Ours) | | Crossformer (2023) | | w/ IF (Ours) | | TimesNet (2023) | | w/ IF (Ours) | |
|---|---|---|---|---|---|---|---|---|---|---|---|---|---|---|---|---|---|---|---|---|---|---|---|---|---|---|---|---|
| Metric | MSE | MAE | MSE | MAE | MSE | MAE | MSE | MAE | MSE | MAE | MSE | MAE | MSE | MAE | MSE | MAE | MSE | MAE | MSE | MAE | MSE | MAE | MSE | MAE | MSE | MAE | MSE | MAE |
| ETTh1 | 0.533 | 0.519 | **0.506** | **0.516** | 0.527 | 0.525 | **0.510** | 0.528 | **0.535** | **0.525** | 0.535 | 0.529 | 0.524 | **0.523** | **0.523** | 0.526 | 0.533 | 0.522 | **0.498** | **0.511** | 0.607 | 0.570 | **0.587** | **0.564** | 0.596 | 0.568 | **0.535** | **0.546** |
| ETTh2 | 0.202 | 0.315 | **0.193** | **0.305** | 0.204 | 0.317 | **0.194** | **0.306** | 0.202 | 0.315 | **0.198** | **0.309** | 0.193 | **0.307** | **0.189** | 0.308 | 0.192 | 0.308 | **0.187** | **0.301** | 0.296 | 0.395 | **0.238** | **0.360** | 0.212 | 0.327 | **0.204** | **0.315** |
| ETTm1 | 0.486 | 0.474 | **0.473** | 0.475 | 0.490 | 0.488 | **0.477** | **0.480** | 0.487 | 0.481 | **0.478** | **0.478** | **0.470** | 0.474 | 0.470 | **0.473** | 0.477 | 0.476 | **0.471** | **0.470** | 0.538 | 0.516 | **0.532** | **0.511** | 0.548 | 0.512 | **0.508** | **0.497** |
| ETTm2 | 0.151 | 0.269 | **0.147** | **0.266** | 0.145 | 0.262 | **0.145** | **0.261** | 0.154 | 0.272 | **0.150** | **0.268** | **0.142** | **0.259** | 0.144 | 0.263 | 0.147 | 0.266 | **0.144** | **0.262** | 0.203 | 0.319 | **0.183** | **0.304** | 0.155 | 0.271 | **0.148** | **0.264** |
| ECL | 0.170 | **0.258** | **0.165** | 0.261 | 0.166 | 0.265 | **0.161** | **0.257** | 0.172 | **0.261** | **0.167** | 0.264 | **0.181** | **0.275** | 0.186 | 0.277 | 0.199 | 0.289 | **0.184** | **0.274** | 0.212 | 0.308 | **0.193** | **0.294** | 0.190 | **0.289** | **0.188** | 0.291 |
| Traffic | 0.433 | 0.286 | **0.429** | 0.287 | **0.480** | **0.297** | 0.629 | 0.343 | **0.434** | **0.292** | 0.497 | 0.333 | 0.639 | 0.326 | **0.573** | **0.315** | 0.489 | 0.317 | **0.470** | **0.308** | 0.575 | 0.326 | **0.546** | 0.336 | 0.614 | **0.336** | 0.593 | 0.359 |
| Weather | 0.236 | 0.261 | **0.229** | 0.270 | 0.221 | 0.254 | **0.221** | 0.260 | 0.237 | 0.261 | **0.225** | **0.261** | 0.230 | 0.281 | **0.227** | **0.262** | 0.229 | 0.262 | **0.224** | **0.259** | 0.242 | 0.305 | 0.253 | 0.313 | 0.241 | 0.270 | **0.231** | **0.269** |
| Exchange | 0.424 | 0.454 | **0.402** | **0.438** | 0.397 | 0.433 | **0.395** | 0.434 | 0.427 | 0.458 | **0.412** | **0.449** | 0.440 | 0.446 | **0.294** | **0.393** | 0.441 | 0.454 | **0.422** | **0.446** | 0.798 | 0.654 | 1.338 | 0.825 | 0.425 | 0.456 | **0.420** | **0.454** |
| ILI | 2.893 | 1.122 | **2.827** | **1.120** | 3.134 | 1.174 | **2.938** | **1.147** | 2.991 | 1.164 | **2.871** | **1.119** | 4.631 | 1.432 | **3.641** | **1.246** | 3.365 | 1.155 | **3.185** | **1.127** | **4.246** | **1.440** | 5.128 | 1.592 | **3.264** | **1.162** | 3.734 | 1.177 |
| PEMS03 | 0.107 | 0.214 | 0.109 | 0.216 | 0.122 | 0.235 | **0.102** | **0.209** | 0.414 | 0.392 | **0.209** | **0.293** | 0.196 | 0.298 | **0.159** | **0.270** | 0.252 | 0.341 | **0.159** | **0.263** | 0.167 | 0.279 | **0.146** | **0.259** | 0.175 | 0.260 | **0.106** | **0.208** |
| PEMS04 | 0.112 | 0.223 | **0.107** | **0.216** | 0.113 | 0.231 | **0.098** | **0.207** | 0.187 | 0.282 | **0.124** | **0.234** | 0.233 | 0.328 | **0.191** | **0.296** | 0.334 | 0.393 | **0.186** | **0.290** | 0.232 | 0.331 | **0.132** | **0.249** | 0.222 | 0.325 | **0.117** | **0.232** |
| PEMS07 | 0.096 | 0.195 | **0.092** | **0.195** | 0.100 | 0.205 | **0.078** | **0.178** | 0.884 | 0.629 | **0.149** | **0.249** | 0.211 | 0.302 | **0.174** | **0.272** | 0.275 | 0.351 | **0.173** | **0.267** | 0.196 | 0.290 | **0.155** | **0.259** | 0.167 | 0.280 | **0.109** | **0.215** |
| PEMS08 | 0.118 | 0.218 | **0.107** | **0.212** | 0.144 | 0.260 | **0.108** | **0.217** | 0.860 | 0.626 | **0.179** | **0.284** | 0.232 | 0.320 | **0.176** | **0.282** | 0.301 | 0.371 | **0.172** | **0.276** | 0.212 | 0.317 | **0.142** | **0.257** | 0.210 | 0.315 | **0.128** | **0.242** |
| Solar | 0.222 | **0.253** | **0.207** | **0.253** | 0.239 | 0.290 | **0.231** | **0.272** | 0.241 | 0.278 | **0.228** | **0.271** | 0.244 | 0.298 | **0.238** | **0.286** | 0.240 | 0.288 | **0.229** | **0.272** | 0.299 | 0.344 | **0.222** | **0.283** | 0.260 | 0.273 | **0.212** | **0.270** |
| Promotion | — | — | 3.71% | 0.60% | — | — | 4.97% | 3.54% | — | — | 18.5% | 11.0% | — | — | 10.4% | 5.12% | — | — | 14.5% | 8.82% | — | — | 6.62% | 4.07% | — | — | 14.3% | 7.22% |

Compared with the initial results, all baselines equipped with IF exhibit consistently improved performance, demonstrating obviously lower MSE/MAE across most of the datasets. This clearly validates the effectiveness and generality of IF, indicating that decoding time series by implicitly predicting and combining constituent components indeed leads to a more accurate forecast. Notably, the performance gains are particularly pronounced on PEMS datasets, in which all models outperform their original results with average decreased MSE and MAE of **30.4%** and **18.2%**. We attribute these substantial improvements to IF's ability to directly access low-frequency waves and to perform separated predictions over long short-term patterns, which facilitates forecasting time series that are mixed with extreme fluctuations and rapid changes, such as PEMS. Furthermore, it is evident from the table that the average improvement of IF in MSE is more significant than that in MAE, revealing that IF produces globally more precise forecasts, and predicting frequency components does offer a more holistic view of the future series. Finally, the experimental results also imply that the encoders of these models are actually capable of extracting useful information from the data, while their original point-wise decoding schemes are insufficient to fully leverage the learned information.

## 4.2   Method Analysis

In this section, we further analyze the proposed Implicit Forecaster. For a fair comparison, all models in the following experiments will adopt the same time series encoder: a standard Transformer encoder with an inverted time series embedding [13].

**Ablation study**   We gradually replace/adjust the proposed module/mechanism, thereby investigating their contribution to the method's overall performance. We consider two ablations: 1) IF: we replace IF with other decoders, including Linear forecaster that is common in recent works [4, 13, 20, 31], MLP forecaster with the same nonlinear process and parameter amount as IF, and the traditional Transformer decoder, 2) Skip-connection: to ensure that the strong performance of IF is not attributed

to its skip-connection, we test the effectiveness of the skip-connection, including disabling it (w/o) and only using it (only, i.e., without using the time series encoder).

Table 2: Ablations of IF. This table presents the standard Transformer's average performance across various forecasting lengths when equipped with different decoders.

| Transformer | Decoder | | | | | | | |
|---|---|---|---|---|---|---|---|---|
| Replace | IF | | Linear | | MLP | | Transformer | |
| Metric | MSE | MAE | MSE | MAE | MSE | MAE | MSE | MAE |
| ETTh2 | **0.188** | **0.301** | 0.201 | 0.313 | 0.199 | 0.311 | 0.202 | 0.316 |
| Exchange | **0.286** | **0.384** | 0.411 | 0.442 | 0.428 | 0.448 | 0.412 | 0.454 |
| ILI | **2.387** | **1.018** | 2.689 | 1.107 | 2.486 | 1.040 | 3.504 | 1.282 |
| PEMS08 | **0.103** | **0.209** | 0.184 | 0.272 | 0.198 | 0.278 | 0.167 | 0.272 |
| Performance | **100%** | **100%** | 77.0% | 88.0% | 77.3% | 88.9% | 73.1% | 84.0% |

Table 3: Ablations of skip-connection. This table details the impact of adjusting the skip-connection in IF.

| IF | Skip-connection | | | | | |
|---|---|---|---|---|---|---|
| Replace | w/ Skip. | | w/o Skip. | | only Skip. | |
| Metric | MSE | MAE | MSE | MAE | MSE | MAE |
| ETTh1 | **0.503** | **0.516** | 0.537 | 0.521 | 0.617 | 0.577 |
| ECL | **0.157** | 0.256 | 0.160 | **0.254** | 0.210 | 0.300 |
| Weather | **0.221** | 0.259 | 0.227 | **0.257** | 0.250 | 0.291 |
| Solar | **0.189** | **0.251** | 0.190 | 0.251 | 0.284 | 0.345 |
| Performance | **100%** | 99.9% | 97.2% | **100%** | 77.8% | 84.0% |

From Table 2, we can clearly witness that changing IF to other decoders results in significant degradation of the model performance. This supports that when processing equivalent input from the encoder, IF is considerably better at transforming high-level time series representations into more precise forecasts, which underscores its substantial effectiveness. As a comparison, particularly on the PEMS dataset, Linear forecaster performs much poorer than IF, revealing that point-wise prediction can struggle in handling long-term patterns. Besides, MLP forecaster is still inferior to IF, which implies that the superiority of IF is not own to nonlinear transformation, but rather to the success of the pattern-separated forecasting method. Notably, the Transformer decoder performs the worst among these forecasters, indicating conventional heavy decoder designs may be overly complicated and thus susceptible to noise interference.

From the results in Table 3, the skip-connection in IF contributes effectively to some of the datasets, in which disabling it only leads to a performance degradation of 2.8% in MSE, indicating its influential but not decisive role in our method's strong performance. Besides, using only skip-connection also shows good forecasting performance, proving that the frequency-domain information in the input time series can associated with future observations, thus can provide effective features for forecasting.

**Computational efficiency** We use the standard big-$\mathcal{O}$ asymptotic notation for analyzing the time complexity of IF. According to the symmetric properties in the frequency domain described in Section 3.2, for a frequency pool size $P$, there will be $\frac{3}{2}P + 3$ output neurons in IF used for making predictions ($\frac{P}{2} + 1$ neurons for predicting amplitudes, sine of phases and cosine of phases, respectively). Therefore, the forward pass complexity of IF is simply bounded by $\mathcal{O}(P)$. However, after predicting frequency-domain features, an iDFT is applied to reconstruct the time signal. The iDFT can be implemented efficiently with a complexity

Table 4: The computational cost of different forecasters across various datasets. The results are measured over the prediction length of $L = 96$ for the PEMS08 dataset and $L = 720$ for the rest of the datasets.

| Decoder | | IF | Linear | MLP | Transformer |
|---|---|---|---|---|---|
| ETTh1 | Speed (ms/iter) | 14.12 | 12.11 | 11.45 | 14.44 |
| | Memory (GB) | 0.241 | 0.190 | 0.217 | 0.251 |
| ECL | Speed (ms/iter) | 65.43 | 59.73 | 64.05 | 68.57 |
| | Memory (GB) | 1.119 | 0.791 | 0.888 | 1.164 |
| Weather | Speed (ms/iter) | 19.23 | 17.51 | 17.76 | 20.43 |
| | Memory (GB) | 0.388 | 0.328 | 0.352 | 0.406 |
| PEMS08 | Speed (ms/iter) | 34.48 | 28.17 | 31.25 | 43.48 |
| | Memory (GB) | 0.899 | 0.538 | 0.637 | 0.940 |

of $\mathcal{O}(P \log P)$ using FFT-based algorithms. Combining both terms, the total time complexity of IF is: prediction complexity + iDFT complexity = $\mathcal{O}(P + P \log P) = \mathcal{O}(P \log P)$. Conventional point-wise forecasters, in contrast, predict future values independently at each time step, with distinct output neurons corresponding to different time points. Hence, their overall computational complexity is $\mathcal{O}(L)$. In experiments, we measured the training speed (ms/iter) and memory footprint (GB) of IF, comparing its computational cost with other forecasters. The results are provided in Table 4, showing that IF has a higher computational cost compared to simpler forecasters such as Linear or MLP, yet it remains more efficient than the Transformer decoder, thus this additional cost does not represent a significant disadvantage.

**Varying lookback length**    The previous methods [13] have verified that treating the entire time series as a token and learning from the variate dimension can benefit the model from a longer lookback window, achieving more accurate forecasting. Considering this can be attributed to the encoder, we believe that the distinct advantages of decoders only emerge with a shorter input. Therefore, we take another perspective that varies the lookback window size from long to short, further validating IF's effectiveness compared to point-wise decoding methods. Figure 3 shows performance changes of Transformer when shortening the input length, where the results surprisingly exhibit that using IF as the decoder not only outperforms the Linear forecaster across all input lengths but that the gap even widens as the input length reduces. This reveals that adopting IF as the decoder could also empower the model's robustness under limited input data, illustrating its unique strength in extracting maximum details from minimal information.

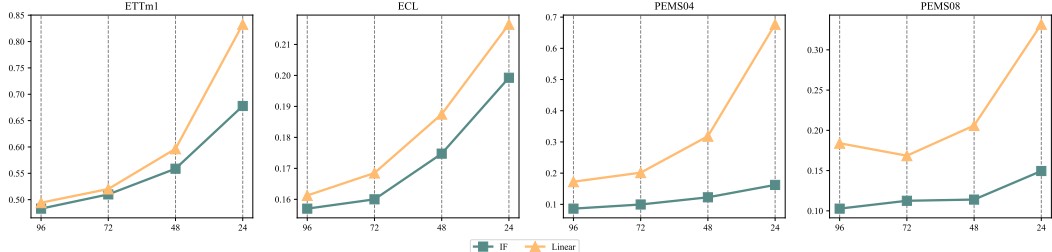

Figure 3: Forecasting performance of Transformer when equipped with Implicit Forecaster (green) and Linear forecaster (yellow) after shortening the lookback window to $T \in \{96, 72, 48, 24\}$. Results are averaged over all prediction lengths, where Implicit Forecaster consistently outperforms Linear forecaster, especially for shorter input lengths.

**Visualization analysis**    Implicitly predicting constituent fluctuations with IF also offers better interpretability. Figure 4 shows a case visualization from the ECL dataset, where the left heat map illustrates the absolute weights learned by AHead for the waves in IF's frequency pool. These higher-weighted frequencies can be regarded as the most influential ones identified across the entire training set. The right heat map displays the corresponding predicted energy distribution (i.e., amplitudes) for these frequencies. We can clearly observe that frequencies with larger weights tend to concentrate higher energy, indicating that IF not only learns which frequencies are important but also actively allocates more representational capacity to them during forecasting. This pattern suggests that IF implicitly captures the dominant periodic components driving the whole temporal dynamics, offering an interpretable decomposition for the forecasting process.

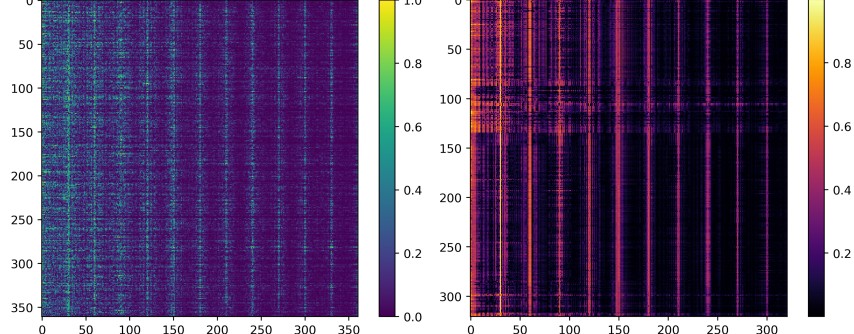

Figure 4: Weights and Prediction Visualisation. The x-axis corresponds to IF's frequency pool.

**How does IF solve the problem of lacking global views in previous methods?**

Frequency components can capture global characteristics of time series, as they are computed from the entire series. This global property has been validated by many forecasting methods and utilized to enhance the model's global views when learning the input [54, 45]. Accordingly, we believe that they can also be used to globally express the output when making predictions. Therefore, the proposed IF module predicts amplitudes and phases of frequency components, and these components are computed as global fluctuations of future series, improving global views of the forecasting phase.

### 4.3 Limitation Analysis

Despite its strong forecasting capabilities, our Implicit Forecaster still exhibits opportunities for further enhancement in terms of efficiency and overall performance. For example, predicting all spectral components may introduce some inefficiencies, as selectively focusing on the more contributive frequencies could potentially reduce computational overhead without significantly compromising the forecasts' accuracy. Besides, employing a fixed-size frequency pool might occasionally be suboptimal, as a manually defined spectrum might not precisely align with the optimal frequencies that best fit the dataset. Moreover, while the Implicit Forecaster is primarily tailored for forecasting tasks, exploring its adaptations across different scenarios could provide additional values. By highlighting these limitations, we aim to encourage further research to refine our method.

## 5    Conclusion

In this paper, we improve the forecasting phase of recently booming TSF methods, which have lacked attention in previous works. We propose Implicit Forecaster, a novel decoding module to replace the output layers of existing forecasting models. It predicts learned historical information into various implicitly represented constituent waves and effectively combines them to form the forecast. Experimental results demonstrate that our Implicit Forecaster can enhance the performance of current mainstream forecasting models, achieving consistent state-of-the-art on multiple real-world datasets. We discover that the forecasting phase of TSF can be an interesting and compelling point of research. Future improvements may include: 1) efficient decoding by predicting key frequencies only, and 2) accurate prediction for localized patterns by incorporating other frequency methods (e.g., Wavelets).

### Acknowledgments and Disclosure of Funding

We would like to acknowledge the support from ARC DP240102088, WIS-MBZUAI 142571, NSFC 623B2002, ARC DE240101089, LP240100101, DP230101540, NSF & CSIRO Responsible AI Program 2303037, the NSF under Award No. 2229881, the AI Institute for Societal Decision Making (AI-SDM), the National Institutes of Health (NIH) under Contract R01HL159805, and grants from Quris AI, Florin Court Capital, MBZUAI-WIS Joint Program, and Al Deira Causal Education Project.

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

# A  Experimental Details

## A.1  Dataset Descriptions

We use 14 real-world datasets to comprehensively evaluate the performance of baseline methods on the long-term time series forecasting task. Detailed descriptions of these datasets are as follows:

**ETT**[2] [53]: The Electricity Transformer Temperature dataset records oil temperatures and multiple power load situations collected from two counties in China between July 2016 and July 2018. It consists of four subsets with different time granularities: ETTh1 and ETTh2 are sampled every hour, while ETTm1 and ETTm2 are sampled every 15 minutes.

**ECL**[3] [53]: The Electricity Consuming Load dataset records the hourly electricity consumption (in kWh) of 321 clients over the period from 2012 to 2014.

**Traffic**[4] [41]: The Traffic dataset records road occupancy rates on San Francisco Bay area freeways, measured hourly by 862 sensors from January 2015 to December 2016, sourced from the California Department of Transportation.

**Weather**[5] [53]: The Weather dataset from the Max Planck Biogeochemistry Institute in Germany records 21 meteorological factors, such as air temperature and humidity. This climatological time series is sampled every 10 minutes throughout the entire year of 2020.

**Exchange Rate** [43]: The Exchange dataset compiles daily exchange rate panel data for eight countries, including Australia, Britain, Canada, Switzerland, China, Japan, New Zealand, and Singapore, covering the period from 1990 to 2016.

**ILI**[6] [25]: The Influenza-Like Illness dataset captures the weekly ratio of influenza-like illness patients to the total number of patients, reported by the Centers for Disease Control and Prevention of the United States from 2002 to 2021.

**PeMS** [12]: This transportation dataset provided by the Caltrans Performance Measurement System records California traffic features like flow, occupancy, and speed. We use four of its public subsets: PEMS03, PEMS04, PEMS07, and PEMS08, each with the same sampling rate of 5 minutes.

**Solar Energy**[7] [10]: The Solar-Energy dataset records solar power production from 137 photovoltaic (PV) plants in Alabama State for the year 2006, with data sampled every 10 minutes.

Table 5: Dataset details. Dim refers to the number of variates in the dataset. Input Length refers to the number of historical timesteps used for making forecasts and Prediction Length denotes to the number of future timesteps to be predicted. Dataset Size denotes the respective timepoints in training, validation and test set. Frequency refers to the sampling interval between each timepoint.

| Dataset | Dim | Input Length | Prediction Length | Dataset Size | Frequency | Domain |
|---|---|---|---|---|---|---|
| ETTh1, ETTh2 | 7 | 96 | $\{96, 192, 336, 720\}$ | (12194, 2613, 2613) | Hourly | Temperature |
| ETTm1, ETTm2 | 7 | 96 | $\{96, 192, 336, 720\}$ | (48776, 10452, 10452) | 15min | Temperature |
| Electricity | 321 | 96 | $\{96, 192, 336, 720\}$ | (18412, 3945, 3947) | Hourly | Electricity |
| Traffic | 862 | 96 | $\{96, 192, 336, 720\}$ | (12280, 2631, 2633) | Hourly | Transportation |
| Weather | 21 | 96 | $\{96, 192, 336, 720\}$ | (36887, 7904, 7905) | 10min | Weather |
| Exchange Rate | 8 | 96 | $\{96, 192, 336, 720\}$ | (5311, 1138, 1139) | Daily | Finance |
| Illness | 7 | 36 | $\{24, 36, 48, 60\}$ | (676, 144, 146) | Weekly | Health |
| PEMS03 | 358 | 96 | $\{12, 24, 48, 96\}$ | (18345, 3931, 3932) | 5min | Transportation |
| PEMS04 | 307 | 96 | $\{12, 24, 48, 96\}$ | (11894, 2548, 2550) | 5min | Transportation |
| PEMS07 | 883 | 96 | $\{12, 24, 48, 96\}$ | (19756, 4233, 4235) | 5min | Transportation |
| PEMS08 | 170 | 96 | $\{12, 24, 48, 96\}$ | (12499, 2678, 2679) | 5min | Transportation |
| Solar-Energy | 137 | 96 | $\{96, 192, 336, 720\}$ | (36792, 7884, 7884) | 10min | Energy |

---

[2] https://github.com/zhouhaoyi/ETDataset
[3] https://archive.ics.uci.edu/ml/datasets/ElectricityLoadDiagrams20112014
[4] http://pems.dot.ca.gov
[5] https://www.bgc-jena.mpg.de/wetter/
[6] https://gis.cdc.gov/grasp/fluview/fluportaldashboard.html
[7] http://www.nrel.gov/grid/solar-power-data.html

## A.2 Baseline Descriptions

We carefully select 7 mainstream TSF models as baselines for comparison, which are representative methods from diverse model types. Detailed descriptions of these baselines are as follows:

**SOFTS** [4]: SOFTS is an MLP-based method that leverages a STar Aggregate-Redistribute (STAR) module to capture the core representation of time series for efficient and accurate forecasting. The core representation is shared across channels and carries global information on all variates. Official implementation of SOFTS is available at this repository: `SOFTS`.

**TimeXer** [38]: TimeXer is a Transformer-based method that can simultaneously leverage endogenous and exogenous variate information. It captures internal information of time series via patch-wise self-attention while integrating external time series information via variate-wise cross-attention, thereby achieving accurate forecasting. Official implementation of TimeXer is available at this repository: `TimeXer`.

**iTransformer** [13]: iTransformer is a Transformer-based method that embeds the entire time series of each variate as a token and inverts the Transformer's role to apply the attention and feed-forward network on the variate dimension. It models series-wise dependencies via multivariate correlating, making it proficient in capturing and learning cross-variate information. Official implementation of iTransformer is available at this repository: `iTransformer`.

**SegRNN** [11]: SegRNN is an RNN-based method that introduces Segment-wise Iterations and Parallel Multi-step Forecasting (PMF) to significantly reduce the required recurrent iterations for forecasting, achieving remarkable performance and inference speed. Official implementation of SegRNN is available at this repository: `SegRNN`.

**PatchTST** [20]: PatchTST is a Transformer-based method that considers channel independence between variates and adopts a Vision Transformer-like strategy [2], segmenting time series into smaller, semantically richer patches. It models patch-wise dependencies, thereby enhancing local information processing for each variate. Official implementation of PatchTST is available at this repository: `PatchTST`.

**Crossformer** [50]: Crossformer is a Transformer-based method that introduces the Dimension-Segment-Wise (DSW) embedding and Two-Stage Attention (TSA) layer. It models patch-wise dependencies across multiple variates, allowing it to effectively learn cross-time and cross-variate information. Official implementation of Crossformer is available at this repository: `Crossformer`.

**TimesNet** [40]: TimesNet is a Frequency-domain integrated CNN-based method that extends 1D time series into 2D space with a Fast Fourier Transform (FFT) for better time series representation. This improves its capability for learning multi-periodicity and extracting complex long short-term temporal patterns. Official implementation of TimesNet is available at this repository: `TimesNet`.

The `Time-Series-Library` provided by TimesNet [40] offers a fair implementation of baseline models. It is built on the source code and configurations provided by each model's original paper.

## A.3 Implementations

All the models and experimental frameworks are implemented entirely in Python and built upon PyTorch 2.0 [22]. All the experiments reported in this paper are conducted on a 16-core AMD EPYC 9654 CPU and a single NVIDIA RTX 4090 GPU. We select Adam [9] as the optimizer and MSE loss to learn the model parameters. The learning rate is scheduled to follow an exponential decay pattern during training, which is halved at the end of each epoch. The number of training epochs is determined using an early stopping strategy, where the training is stopped when the model's performance (i.e., loss) ceases to improve on the validation set for a maximum of 3 times.

For experimental fairness, throughout the experiments, we did not perform any hyperparameter tuning for any of the baseline models or models equipped with IF. Instead, to ensure consistency and accurately assess our approach, all models used the same hyperparameters before and after applying IF. All baseline methods and their hyperparameters, scripts, and experimental frameworks are strictly followed by the `Time-Series-Library` repository provided by TimesNet [40], which is a comprehensive and fair platform for time series analysis.

**Loss Function**   In our experiments, all models are trained by minimizing the MSE loss to optimize their model parameters. Given the predicted future time series $\hat{\mathbf{Y}} \in \mathbb{R}^{L \times N}$ and the corresponding ground truth $\mathbf{Y} \in \mathbb{R}^{L \times N}$, where $L$ denotes the forecasting horizon and $N$ represents the number of variates, the MSE loss between $\hat{\mathbf{Y}}$ and $\mathbf{Y}$ is defined as follows:

$$\mathcal{L}_{\mathrm{MSE}}(\hat{\mathbf{Y}}, \mathbf{Y}) = \frac{1}{LN} \sum_{i=1}^{L} \sum_{j=1}^{N} \left( \hat{\mathbf{Y}}_{i,j} - \mathbf{Y}_{i,j} \right)^2, \tag{7}$$

where $\hat{\mathbf{Y}}_{i,j}, \mathbf{Y}_{i,j} \in \mathbb{R}$ are the value of the time series at the $i$-th time step for the $j$-th variate.

**Evaluation Metrics**   To align with previous works , we evaluate the performance of models with two metrics: the Mean Squared Error (MSE) and the Mean Absolute Error (MAE). In multivariate time series forecasting, given the predicted series $\hat{\mathbf{Y}}$ and the true series $\mathbf{Y}$,

**Mean Squared Error (MSE):**

$$\mathrm{MSE}(\hat{\mathbf{Y}}, \mathbf{Y}) = \frac{1}{LN} \sum_{i=1}^{L} \sum_{j=1}^{N} \left( \hat{\mathbf{Y}}_{i,j} - \mathbf{Y}_{i,j} \right)^2, \tag{8}$$

**Mean Absolute Error (MAE):**

$$\mathrm{MAE}(\hat{\mathbf{Y}}, \mathbf{Y}) = \frac{1}{LN} \sum_{i=1}^{L} \sum_{j=1}^{N} \left| \hat{\mathbf{Y}}_{i,j} - \mathbf{Y}_{i,j} \right|. \tag{9}$$

# B   Algorithmic Details

We provide a pseudocode of IF's algorithmic details, clearly demonstrating its step-by-step process for time series forecasting. Note that: 1) $\mathrm{MLP}_{\{amp, sin, cos\}} : \mathbb{R}^{D+T} \mapsto \mathbb{R}^P$ are nonlinear neural networks with two dense layers, which are applied to the last dimension of the input, 2) Concatenate is the concatenation operation that concatenates tensors at their last dimension.

---

**Algorithm 1** Implicit Forecaster

---

**Define:** input length $T$; output length $L$; variates number $N$; model dimension $D$; spectrum size $P$
**Require:** encoder representation $\mathbf{X}_{enc} \in \mathbb{R}^{N \times D}$; input time series $\mathbf{X} \in \mathbb{R}^{T \times N}$
**Ensure:** output time series $\hat{\mathbf{Y}} \in \mathbb{R}^{L \times N}$

1: ▷ Discrete Fourier Transform the input time series, taking amplitudes and phases.
2: $\mathbf{S} = \mathcal{F}(\mathbf{X}^\top)$                                                         ▷ $\mathbf{S} \in \mathbb{C}^{N \times T}$
3: $\mathbf{A} = |\mathbf{S}|$                                                    ▷ $\mathbf{A} \in \mathbb{R}_{\geq 0}^{N \times T}$
4: $\phi = \arg(\mathbf{S})$                                          ▷ $\phi \in [-\pi, \pi]^{N \times T}$
5: ▷ Run Amplitude Head, predict amplitudes.
6: $\hat{\mathbf{A}} = |\mathrm{MLP}_{amp}(\mathrm{Concatenate}(\mathbf{X}_{enc}, \mathbf{A}))|$            ▷ $\hat{\mathbf{A}} \in \mathbb{R}_{\geq 0}^{N \times P}$
7: ▷ Run Phase Head, predict phases.
8: $\hat{\boldsymbol{\alpha}} = \mathrm{Tanh}(\mathrm{MLP}_{sin}(\mathrm{Concatenate}(\mathbf{X}_{enc}, \phi)))$        ▷ $\hat{\boldsymbol{\alpha}} \in [-1, 1]^{N \times P}$
9: $\hat{\boldsymbol{\beta}} = \mathrm{Tanh}(\mathrm{MLP}_{cos}(\mathrm{Concatenate}(\mathbf{X}_{enc}, \phi)))$        ▷ $\hat{\boldsymbol{\beta}} \in [-1, 1]^{N \times P}$
10: $\hat{\phi} = \mathrm{atan2}(\hat{\boldsymbol{\alpha}}, \hat{\boldsymbol{\beta}})$                                    ▷ $\hat{\phi} \in [-\pi, \pi]^{N \times P}$
11: ▷ Combine predicted waves through inverse Discrete Fourier Transform.
12: $\hat{\mathbf{Y}} = \mathcal{F}^{-1}(\hat{\mathbf{A}} \cdot e^{j\hat{\phi}})$                                 ▷ $\hat{\mathbf{Y}} \in \mathbb{R}^{N \times P}$
13: ▷ Take the part within the output length.
14: $\hat{\mathbf{Y}} = \mathrm{Crop}_{:L}(\hat{\mathbf{Y}}^\top)$                                ▷ $\hat{\mathbf{Y}} \in \mathbb{R}^{L \times N}$
15: **return** $\hat{\mathbf{Y}}$

---

## C   Statistical Significance Test

**Error Bar**   We evaluate the robustness of IF across different random seeds using six datasets. Table 6 reports the standard deviations of the Transformer's performance when equipped with IF over five random runs, demonstrating that IF yields consistently stable results.

Table 6: This table presents the performance robustness when the standard Transformer is equipped with IF. The results are obtained from 5 random runs.

| Dataset | ETTh2 | | ECL | | Weather | |
|---|---|---|---|---|---|---|
| Horizon | MSE | MAE | MSE | MAE | MSE | MAE |
| 96 | $0.152\pm 0.000$ | $0.272\pm 0.002$ | $0.132\pm 0.000$ | $0.229\pm 0.000$ | $0.151\pm 0.002$ | $0.199\pm 0.001$ |
| 192 | $0.181\pm 0.002$ | $0.296\pm 0.002$ | $0.147\pm 0.001$ | $0.244\pm 0.000$ | $0.197\pm 0.001$ | $0.244\pm 0.002$ |
| 336 | $0.195\pm 0.002$ | $0.307\pm 0.002$ | $0.161\pm 0.001$ | $0.262\pm 0.001$ | $0.237\pm 0.004$ | $0.274\pm 0.003$ |
| 720 | $0.223\pm 0.005$ | $0.328\pm 0.003$ | $0.188\pm 0.003$ | $0.289\pm 0.004$ | $0.301\pm 0.001$ | $0.319\pm 0.003$ |

| Dataset | PEMS03 | | PEMS07 | | PEMS08 | |
|---|---|---|---|---|---|---|
| Horizon | MSE | MAE | MSE | MAE | MSE | MAE |
| 12 | $0.058\pm 0.001$ | $0.160\pm 0.001$ | $0.052\pm 0.001$ | $0.144\pm 0.001$ | $0.062\pm 0.000$ | $0.163\pm 0.000$ |
| 24 | $0.074\pm 0.002$ | $0.181\pm 0.002$ | $0.063\pm 0.001$ | $0.157\pm 0.001$ | $0.077\pm 0.001$ | $0.183\pm 0.001$ |
| 48 | $0.100\pm 0.002$ | $0.209\pm 0.002$ | $0.084\pm 0.002$ | $0.185\pm 0.002$ | $0.107\pm 0.002$ | $0.218\pm 0.002$ |
| 96 | $0.145\pm 0.004$ | $0.259\pm 0.003$ | $0.120\pm 0.002$ | $0.221\pm 0.002$ | $0.165\pm 0.002$ | $0.273\pm 0.002$ |

**Paired t-test**   To verify whether the improvement of the model by IF is statistically significant, we have also repeated the experiments across random runs and conducted a paired t-test to further validate the significance of the performance gain. Specifically, we applied IF to the Transformer backbone while keeping all hyperparameters identical (including the random seed) before and after applying IF to ensure a strictly fair comparison. For the paired t-test, we test the hypothesis that the Transformer equipped with IF achieves a significantly lower MSE than without IF. The experimental results averaged over 4 prediction lengths are as follows:

Table 7: Performance comparison of the standard Transformer without and with IF. Significance levels: * $p < 0.05$, ** $p < 0.01$, *** $p < 0.001$, n.s. **not significant**.

| Transformer (MSE) | Original | w/ IF | p-value | Significance |
|---|---|---|---|---|
| ETTh1 | $0.539 \pm 0.001$ | $\mathbf{0.504 \pm 0.002}$ | $1.98 \times 10^{-5}$ | *** |
| ETTh2 | $0.200 \pm 0.001$ | $\mathbf{0.188 \pm 0.001}$ | $4.68 \times 10^{-4}$ | *** |
| ETTm1 | $0.508 \pm 0.001$ | $\mathbf{0.493 \pm 0.003}$ | $1.04 \times 10^{-3}$ | ** |
| ETTm2 | $0.155 \pm 0.001$ | $\mathbf{0.148 \pm 0.001}$ | $1.23 \times 10^{-5}$ | *** |
| ECL | $0.162 \pm 0.001$ | $\mathbf{0.159 \pm 0.000}$ | $5.71 \times 10^{-3}$ | ** |
| Traffic | $0.620 \pm 0.001$ | $\mathbf{0.550 \pm 0.002}$ | $7.85 \times 10^{-4}$ | *** |
| Weather | $0.228 \pm 0.000$ | $\mathbf{0.220 \pm 0.000}$ | $2.15 \times 10^{-5}$ | *** |
| Exchange | $0.406 \pm 0.004$ | $\mathbf{0.297 \pm 0.003}$ | $1.31 \times 10^{-5}$ | *** |
| ILI | $2.985 \pm 0.013$ | $\mathbf{2.770 \pm 0.053}$ | $1.03 \times 10^{-2}$ | * |
| PEMS03 | $0.123 \pm 0.001$ | $\mathbf{0.095 \pm 0.000}$ | $9.71 \times 10^{-6}$ | *** |
| PEMS04 | $0.152 \pm 0.022$ | $\mathbf{0.099 \pm 0.006}$ | $3.67 \times 10^{-2}$ | * |
| PEMS07 | $0.103 \pm 0.001$ | $\mathbf{0.081 \pm 0.001}$ | $6.70 \times 10^{-6}$ | *** |
| PEMS08 | $0.176 \pm 0.011$ | $\mathbf{0.104 \pm 0.002}$ | $1.67 \times 10^{-3}$ | ** |
| Solar | $\mathbf{0.188 \pm 0.001}$ | $0.189 \pm 0.001$ | $5.44 \times 10^{-1}$ | n.s. |

From Table 7, it is evident that the performance of our method is considerably robust, and the proposed IF module is stably and statistically significantly outperforms the Transformer's original MLP-based point-wise prediction method (in 13 out of 14 datasets).

# D Showcases

## D.1 Model Prediction Visualisations

We provide visualisations of model forecasts as supplementary showcases, offering a clearer comparison between baseline models and their enhanced versions with the IF decoder. The left column shows the predictions from the baseline models, while the right column shows their predictions after incorporating with IF. The forecasts from each model on the ECL and PEMS03 datasets are illustrated in Figure 5–6, where the visual comparisons clearly show that incorporating IF leads to more accurate and aligned predictions.

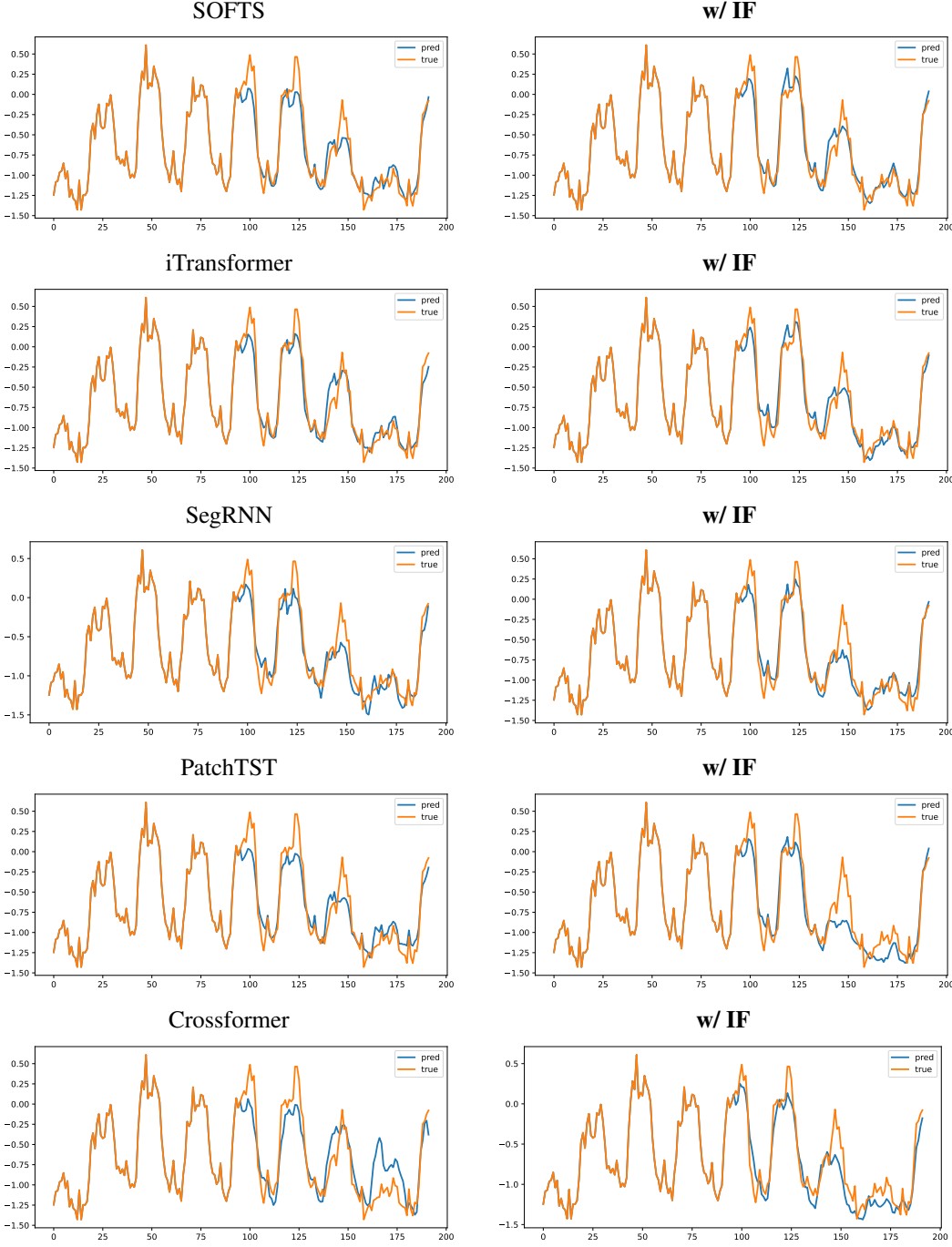

Figure 5: Visualization of input-96-output-96 results on the ECL dataset

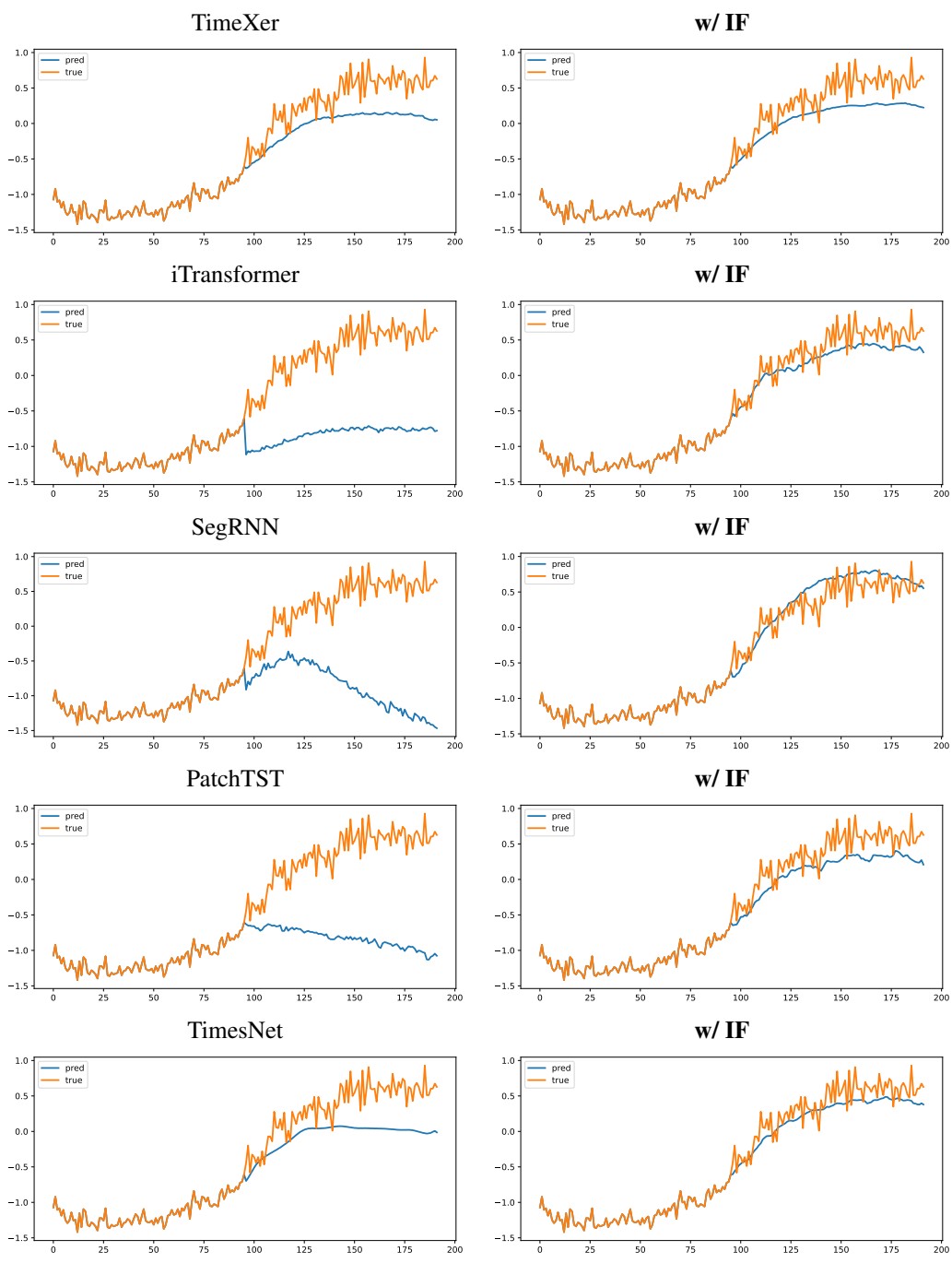

Figure 6: Visualization of input-96-output-96 results on the PEMS03 dataset

### D.2 Decoder Prediction Visualisations

We further provide forecast visualisations of the Transformer equipped with different decoders (i.e., Implicit Forecaster, Linear Forecaster, MLP Forecaster, and Transformer Decoder). The results on the ETTh2 and ILI datasets are shown in Figure 7–8, where IF produces visibly more accurate and smoother predictions compared to the other decoders.

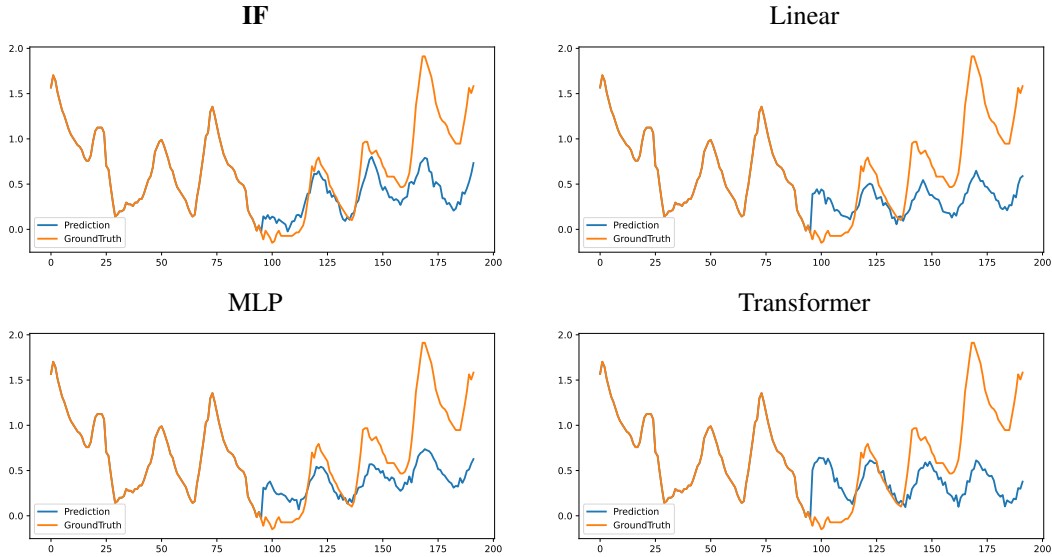

Figure 7: Visualization of input-96-predict-96 results on the ETTh2 dataset

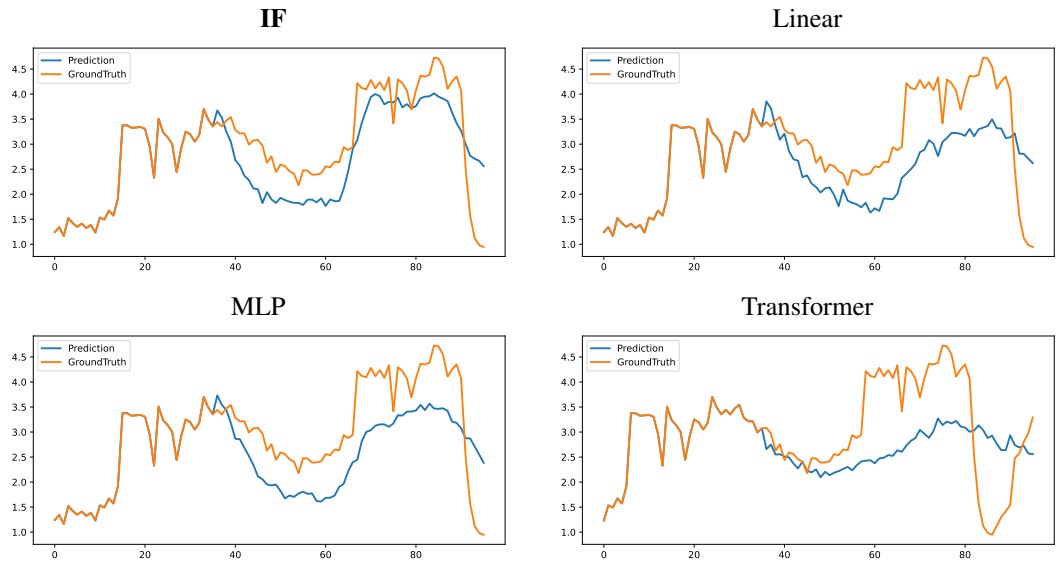

Figure 8: Visualization of input-36-predict-60 results on the ILI dataset

## E    Full Results

Table 8 provides the complete multivariate time series forecasting results of all baseline models across 14 datasets, along with their enhanced versions equipped with the proposed Implicit Forecaster (**w/ IF**). The results show that IF brings consistent and substantial performance improvements in most

cases. The more pronounced gains in MSE compared to MAE also indicate that IF enhances the model's global views of the future time series, leading to more accurate forecasts.

Table 8: The comprehensive results of multivariate time series forecasting. The lookback window length is set to $T = 36$ for the ILI and $T = 96$ for the others. Avg means the average result from all four prediction lengths.

| Models | | SOFTS (2024) | | w/ IF (Ours) | | TimeXer (2024) | | w/ IF (Ours) | | iTransformer (2024) | | w/ IF (Ours) | | SegRNN (2023) | | w/ IF (Ours) | | PatchTST (2023) | | w/ IF (Ours) | | Crossformer (2023) | | w/ IF (Ours) | | TimesNet (2023) | | w/ IF (Ours) | |
|---|---|---|---|---|---|---|---|---|---|---|---|---|---|---|---|---|---|---|---|---|---|---|---|---|---|---|---|---|---|
| Metric | | MSE | MAE | MSE | MAE | MSE | MAE | MSE | MAE | MSE | MAE | MSE | MAE | MSE | MAE | MSE | MAE | MSE | MAE | MSE | MAE | MSE | MAE | MSE | MAE | MSE | MAE | MSE | MAE |
| ETTh1 | 96 | 0.442 | 0.461 | **0.426** | **0.458** | 0.423 | 0.457 | **0.420** | 0.461 | **0.447** | **0.469** | 0.456 | 0.474 | **0.432** | **0.463** | 0.436 | 0.467 | 0.425 | 0.457 | **0.411** | **0.453** | 0.437 | 0.476 | 0.437 | **0.474** | 0.484 | 0.501 | **0.433** | **0.475** |
| | 192 | 0.493 | 0.495 | **0.469** | **0.489** | 0.487 | 0.503 | **0.461** | **0.494** | 0.495 | 0.503 | 0.490 | 0.503 | 0.480 | 0.497 | **0.467** | **0.490** | 0.477 | 0.494 | 0.485 | 0.500 | 0.487 | 0.505 | 0.581 | 0.563 | 0.542 | 0.536 | **0.498** | **0.521** |
| | 336 | 0.529 | **0.519** | **0.510** | 0.523 | 0.529 | 0.528 | 0.524 | 0.541 | 0.537 | 0.526 | 0.533 | 0.532 | 0.524 | 0.526 | **0.502** | **0.517** | 0.552 | 0.530 | **0.497** | **0.511** | 0.640 | 0.602 | 0.644 | 0.590 | 0.606 | 0.576 | **0.548** | 0.553 |
| | 720 | 0.667 | 0.602 | **0.618** | **0.595** | 0.668 | **0.612** | 0.637 | 0.618 | 0.662 | 0.601 | 0.660 | 0.608 | **0.659** | 0.608 | 0.689 | 0.631 | 0.678 | 0.607 | **0.601** | **0.582** | 0.862 | 0.696 | **0.684** | 0.630 | 0.752 | 0.657 | **0.662** | 0.637 |
| | Avg | 0.533 | 0.519 | **0.506** | **0.516** | 0.527 | 0.525 | **0.510** | 0.528 | 0.535 | 0.525 | 0.535 | 0.529 | 0.524 | **0.523** | **0.523** | 0.526 | 0.533 | 0.522 | **0.498** | **0.511** | 0.607 | 0.570 | **0.587** | **0.564** | 0.596 | 0.568 | **0.535** | **0.546** |
| ETTh2 | 96 | 0.159 | 0.279 | **0.154** | **0.273** | 0.153 | 0.273 | **0.151** | **0.270** | 0.160 | 0.280 | **0.159** | **0.277** | 0.149 | 0.269 | 0.148 | 0.270 | 0.152 | 0.272 | **0.147** | **0.266** | 0.179 | 0.310 | **0.171** | **0.302** | 0.168 | 0.290 | **0.161** | **0.279** |
| | 192 | 0.188 | 0.304 | **0.182** | **0.295** | 0.184 | 0.301 | **0.177** | **0.292** | 0.187 | 0.303 | **0.186** | **0.300** | 0.180 | 0.296 | **0.175** | **0.295** | 0.183 | 0.302 | **0.179** | **0.295** | 0.261 | 0.381 | **0.223** | **0.353** | 0.209 | 0.323 | **0.199** | **0.309** |
| | 336 | 0.208 | 0.320 | **0.195** | **0.309** | 0.215 | 0.327 | **0.209** | **0.322** | 0.208 | 0.322 | **0.202** | **0.315** | 0.200 | 0.315 | **0.194** | **0.309** | 0.198 | 0.316 | **0.194** | **0.309** | 0.256 | 0.384 | **0.251** | **0.371** | 0.216 | 0.332 | **0.210** | **0.322** |
| | 720 | 0.254 | 0.356 | **0.240** | **0.342** | 0.263 | 0.367 | **0.239** | **0.341** | 0.253 | 0.356 | **0.244** | **0.346** | 0.242 | 0.350 | 0.243 | 0.356 | 0.234 | 0.343 | **0.227** | **0.333** | 0.487 | 0.505 | **0.306** | **0.413** | 0.257 | 0.361 | **0.246** | **0.349** |
| | Avg | 0.202 | 0.315 | **0.193** | **0.305** | 0.204 | 0.317 | **0.194** | **0.306** | 0.202 | 0.315 | **0.198** | **0.309** | 0.193 | 0.307 | **0.189** | 0.308 | 0.192 | 0.308 | **0.187** | **0.301** | 0.296 | 0.395 | **0.238** | **0.360** | 0.212 | 0.327 | **0.204** | **0.315** |
| ETTm1 | 96 | **0.398** | **0.416** | 0.406 | 0.425 | 0.417 | 0.432 | 0.435 | 0.438 | 0.392 | 0.426 | **0.396** | **0.425** | 0.400 | 0.424 | **0.389** | **0.416** | 0.390 | 0.418 | 0.403 | 0.417 | 0.497 | 0.488 | **0.487** | **0.467** | 0.487 | 0.469 | **0.463** | **0.460** |
| | 192 | **0.438** | **0.448** | 0.441 | 0.454 | 0.458 | 0.469 | **0.446** | **0.458** | 0.447 | 0.458 | **0.443** | **0.458** | 0.438 | 0.455 | **0.435** | **0.450** | 0.445 | 0.455 | 0.445 | 0.452 | 0.515 | 0.497 | **0.479** | **0.480** | 0.514 | 0.490 | **0.478** | **0.476** |
| | 336 | 0.520 | 0.492 | **0.488** | **0.485** | 0.509 | 0.503 | **0.495** | **0.495** | 0.523 | 0.498 | **0.499** | **0.493** | 0.492 | 0.490 | **0.490** | **0.483** | 0.541 | 0.524 | **0.531** | **0.517** | 0.573 | 0.534 | **0.517** | **0.505** | 0.572 | 0.547 | **0.517** | **0.505** |
| | 720 | 0.589 | 0.540 | **0.555** | **0.537** | 0.577 | 0.550 | **0.534** | **0.529** | 0.585 | 0.540 | **0.572** | **0.538** | 0.548 | 0.529 | 0.562 | 0.536 | 0.571 | 0.540 | **0.544** | **0.527** | 0.600 | 0.554 | 0.633 | 0.580 | 0.620 | 0.557 | **0.572** | **0.547** |
| | Avg | 0.486 | 0.474 | **0.473** | 0.475 | 0.490 | 0.488 | **0.477** | **0.480** | 0.487 | 0.481 | **0.478** | **0.478** | 0.470 | 0.474 | **0.470** | **0.473** | 0.477 | 0.476 | **0.471** | **0.470** | 0.538 | 0.516 | **0.532** | **0.511** | 0.548 | 0.512 | **0.508** | **0.497** |
| ETTm2 | 96 | 0.109 | 0.227 | **0.101** | **0.220** | **0.098** | 0.217 | 0.099 | **0.217** | 0.107 | 0.227 | **0.105** | **0.225** | 0.097 | 0.216 | 0.100 | 0.217 | 0.102 | 0.222 | **0.100** | **0.219** | 0.112 | 0.240 | **0.105** | **0.227** | 0.106 | 0.225 | **0.101** | **0.218** |
| | 192 | 0.130 | 0.252 | **0.128** | **0.249** | 0.125 | 0.247 | **0.124** | **0.244** | 0.132 | 0.255 | **0.130** | **0.253** | 0.123 | 0.244 | 0.125 | 0.246 | 0.128 | 0.251 | **0.127** | **0.250** | 0.147 | 0.267 | **0.137** | **0.262** | 0.136 | 0.254 | **0.130** | **0.248** |
| | 336 | 0.162 | 0.280 | **0.158** | **0.279** | **0.155** | 0.273 | 0.156 | **0.273** | 0.168 | 0.286 | **0.161** | **0.280** | 0.151 | 0.270 | 0.154 | 0.274 | 0.158 | 0.279 | **0.152** | **0.272** | 0.238 | 0.344 | **0.209** | **0.338** | 0.165 | 0.281 | **0.158** | **0.276** |
| | 720 | 0.204 | 0.315 | **0.202** | **0.314** | 0.201 | 0.311 | **0.200** | **0.309** | 0.209 | 0.319 | **0.205** | **0.315** | 0.195 | 0.306 | 0.198 | 0.313 | 0.194 | 0.309 | **0.195** | **0.308** | 0.315 | 0.425 | **0.282** | **0.391** | 0.215 | 0.322 | **0.203** | **0.313** |
| | Avg | 0.151 | 0.269 | **0.147** | **0.266** | 0.145 | 0.262 | **0.145** | **0.261** | 0.154 | 0.272 | **0.150** | **0.268** | 0.142 | 0.259 | 0.144 | 0.263 | 0.147 | 0.266 | **0.144** | **0.262** | 0.203 | 0.319 | **0.183** | **0.304** | 0.155 | 0.271 | **0.148** | **0.264** |
| ECL | 96 | 0.146 | 0.233 | **0.141** | **0.233** | 0.142 | 0.242 | **0.137** | **0.231** | 0.150 | 0.239 | **0.145** | **0.239** | 0.157 | 0.249 | 0.165 | 0.254 | 0.179 | 0.271 | **0.163** | **0.250** | 0.164 | 0.266 | **0.161** | **0.263** | 0.167 | 0.268 | **0.160** | **0.265** |
| | 192 | 0.157 | 0.245 | **0.157** | **0.250** | 0.155 | 0.252 | **0.151** | **0.247** | 0.160 | 0.248 | **0.158** | **0.253** | 0.169 | 0.260 | 0.171 | 0.263 | 0.184 | 0.275 | **0.170** | **0.259** | 0.176 | 0.277 | **0.182** | **0.277** | 0.182 | 0.282 | **0.178** | **0.281** |
| | 336 | 0.170 | 0.259 | **0.166** | **0.263** | 0.171 | 0.271 | **0.165** | **0.261** | 0.172 | 0.263 | **0.169** | **0.268** | 0.181 | 0.277 | 0.185 | 0.278 | 0.197 | 0.289 | **0.182** | **0.274** | 0.213 | 0.308 | **0.200** | **0.303** | 0.192 | 0.293 | 0.194 | 0.298 |
| | 720 | 0.208 | 0.294 | **0.197** | **0.296** | 0.198 | 0.295 | **0.190** | **0.290** | 0.208 | 0.295 | **0.196** | **0.295** | 0.216 | 0.312 | 0.219 | 0.312 | 0.231 | 0.321 | **0.221** | **0.314** | 0.285 | 0.376 | **0.236** | **0.335** | 0.218 | 0.315 | **0.218** | **0.320** |
| | Avg | 0.170 | 0.258 | **0.165** | **0.261** | 0.166 | 0.265 | **0.161** | **0.257** | 0.172 | 0.261 | **0.167** | **0.264** | 0.181 | 0.275 | 0.184 | 0.277 | 0.199 | 0.289 | **0.184** | **0.274** | 0.212 | 0.308 | **0.193** | **0.294** | 0.190 | 0.289 | **0.188** | 0.291 |
| Traffic | 96 | 0.393 | 0.267 | **0.390** | **0.267** | 0.438 | 0.278 | 0.690 | 0.341 | **0.401** | 0.277 | 0.432 | 0.302 | 0.609 | 0.313 | **0.530** | 0.298 | 0.463 | 0.304 | **0.439** | **0.293** | 0.544 | 0.311 | **0.512** | 0.319 | 0.579 | **0.322** | 0.573 | 0.348 |
| | 192 | 0.426 | 0.281 | **0.420** | **0.280** | 0.471 | 0.296 | 0.664 | 0.347 | **0.427** | 0.288 | 0.458 | 0.311 | 0.626 | 0.319 | **0.559** | 0.306 | 0.478 | 0.311 | **0.460** | **0.302** | 0.565 | 0.320 | **0.525** | 0.329 | 0.608 | **0.332** | 0.584 | 0.363 |
| | 336 | 0.444 | 0.289 | **0.440** | **0.291** | 0.487 | 0.298 | 0.621 | 0.347 | **0.441** | 0.294 | 0.482 | 0.324 | 0.646 | 0.326 | **0.571** | 0.316 | 0.493 | 0.317 | **0.475** | **0.308** | 0.579 | 0.328 | **0.548** | 0.337 | 0.622 | **0.338** | 0.592 | 0.350 |
| | 720 | 0.468 | 0.307 | **0.467** | **0.309** | 0.523 | 0.316 | 0.541 | 0.338 | **0.467** | 0.310 | 0.617 | 0.396 | 0.675 | 0.345 | **0.633** | 0.339 | 0.520 | 0.335 | **0.507** | **0.331** | 0.610 | 0.346 | **0.598** | 0.358 | 0.648 | **0.351** | 0.623 | 0.376 |
| | Avg | 0.433 | **0.286** | 0.429 | 0.287 | 0.480 | 0.297 | 0.629 | 0.343 | **0.434** | 0.292 | 0.497 | 0.333 | 0.639 | 0.326 | **0.573** | 0.315 | 0.489 | 0.317 | **0.470** | **0.308** | 0.575 | 0.326 | **0.546** | 0.336 | 0.614 | **0.336** | 0.593 | 0.359 |
| Weather | 96 | 0.164 | 0.205 | **0.161** | 0.214 | 0.147 | 0.194 | 0.145 | 0.196 | 0.164 | 0.202 | **0.154** | **0.201** | 0.166 | 0.218 | **0.205** | 0.208 | 0.163 | 0.230 | 0.164 | 0.233 | 0.165 | 0.213 | 0.156 | 0.210 |
| | 192 | **0.204** | **0.240** | 0.207 | 0.257 | 0.191 | 0.234 | 0.189 | 0.238 | 0.207 | 0.240 | **0.198** | **0.242** | 0.198 | 0.258 | 0.199 | 0.255 | 0.207 | 0.242 | **0.196** | **0.242** | 0.222 | 0.295 | 0.224 | 0.292 | 0.207 | 0.249 | **0.203** | 0.250 |
| | 336 | 0.257 | 0.278 | **0.241** | **0.277** | 0.244 | 0.273 | 0.247 | 0.282 | 0.259 | 0.280 | **0.244** | **0.277** | 0.259 | 0.291 | 0.249 | 0.294 | 0.258 | 0.279 | **0.249** | **0.278** | 0.256 | 0.320 | 0.263 | 0.323 | 0.272 | 0.293 | **0.255** | **0.289** |
| | 720 | 0.319 | 0.322 | **0.309** | **0.331** | 0.304 | 0.314 | 0.303 | 0.324 | 0.319 | 0.322 | **0.305** | **0.323** | 0.317 | 0.348 | 0.305 | 0.340 | 0.320 | 0.320 | 0.312 | 0.321 | 0.327 | 0.377 | 0.359 | 0.404 | 0.322 | 0.327 | **0.309** | 0.326 |
| | Avg | 0.236 | **0.261** | 0.229 | 0.270 | 0.221 | 0.254 | 0.221 | 0.260 | 0.237 | 0.261 | **0.225** | 0.261 | 0.230 | 0.281 | 0.227 | 0.277 | 0.237 | 0.262 | 0.229 | 0.262 | 0.242 | 0.305 | 0.253 | 0.313 | 0.241 | 0.270 | **0.231** | 0.269 |
| Exchange | 96 | 0.089 | 0.214 | **0.082** | **0.204** | 0.086 | 0.207 | **0.082** | **0.204** | 0.103 | 0.231 | **0.102** | **0.229** | 0.095 | 0.208 | 0.091 | 0.217 | 0.112 | 0.240 | **0.086** | **0.206** | 0.236 | 0.352 | 0.262 | 0.372 | 0.096 | 0.225 | **0.090** | **0.217** |
| | 192 | 0.200 | 0.328 | **0.177** | **0.305** | 0.187 | 0.312 | **0.177** | **0.306** | 0.206 | 0.334 | **0.188** | **0.318** | 0.174 | 0.298 | 0.207 | 0.333 | 0.184 | 0.310 | **0.183** | **0.309** | 0.427 | 0.484 | 0.526 | 0.539 | **0.185** | 0.319 | 0.187 | 0.321 |
| | 336 | 0.389 | 0.465 | **0.349** | **0.423** | 0.371 | 0.449 | 0.373 | 0.452 | **0.362** | 0.448 | 0.364 | 0.450 | 0.360 | 0.437 | **0.306** | **0.424** | 0.361 | 0.445 | **0.345** | **0.434** | 1.063 | 0.807 | 2.276 | 1.191 | **0.381** | 0.456 | 0.382 | 0.460 |
| | 720 | 1.017 | 0.809 | **0.999** | **0.807** | 0.943 | 0.763 | 0.950 | 0.773 | 1.038 | 0.820 | **0.994** | **0.798** | 1.142 | 0.848 | 0.578 | 0.608 | 1.128 | 0.847 | **1.074** | **0.834** | 1.466 | 0.973 | 2.287 | 1.198 | 1.037 | 0.825 | **1.023** | 0.816 |
| | Avg | 0.424 | 0.454 | **0.402** | **0.438** | 0.397 | 0.433 | 0.395 | 0.434 | 0.427 | 0.458 | **0.412** | **0.449** | 0.440 | 0.446 | 0.294 | 0.393 | 0.441 | 0.454 | **0.422** | **0.446** | 0.798 | 0.654 | 1.338 | 0.825 | 0.425 | 0.456 | **0.420** | 0.454 |
| ILI | 96 | 2.731 | **1.001** | **2.621** | 1.035 | **2.903** | 1.134 | 2.941 | 1.124 | 2.820 | 1.085 | **2.543** | **1.014** | 3.974 | 1.272 | **2.920** | 1.078 | 2.585 | 1.005 | **2.292** | **1.003** | 3.922 | 1.375 | 4.598 | 1.508 | **2.715** | 1.054 | 2.811 | 1.057 |
| | 192 | **2.665** | **1.105** | 2.837 | 1.121 | 3.052 | 1.153 | **2.677** | **1.119** | 3.034 | 1.173 | **2.880** | **1.113** | 4.366 | 1.379 | **3.167** | **1.173** | 4.506 | 1.292 | **4.131** | **1.252** | 3.971 | 1.383 | 5.034 | 1.573 | **3.670** | **1.245** | 5.866 | 1.333 |
| | 336 | 3.485 | 1.280 | **3.212** | **1.212** | 3.402 | 1.235 | **2.926** | **1.154** | 3.158 | 1.224 | 3.263 | 1.232 | 4.425 | 1.427 | **3.932** | **1.339** | 3.578 | 1.219 | **3.736** | **1.202** | 3.958 | 1.420 | 4.861 | 1.543 | 3.633 | 1.229 | **3.537** | **1.228** |
| | 720 | 2.691 | 1.101 | **2.637** | 1.111 | 3.179 | 1.174 | **3.179** | **1.174** | 2.953 | 1.175 | **2.797** | **1.116** | 5.759 | 1.651 | **4.342** | 1.395 | 2.792 | 1.102 | **2.581** | **1.050** | 5.132 | 1.582 | 6.021 | 1.745 | 3.037 | 1.120 | **2.723** | **1.089** |
| | Avg | 2.893 | 1.122 | **2.827** | **1.120** | 3.134 | 1.174 | **2.938** | **1.147** | 2.991 | 1.164 | **2.871** | **1.119** | 4.631 | 1.432 | **3.641** | **1.246** | 3.365 | 1.155 | **3.185** | **1.127** | 4.246 | 1.440 | 5.128 | 1.592 | **3.264** | **1.162** | 3.734 | 1.177 |
| PEMS03 | 12 | 0.064 | 0.167 | **0.061** | **0.164** | 0.064 | 0.177 | **0.058** | **0.165** | 0.065 | 0.172 | **0.062** | **0.169** | 0.078 | 0.191 | **0.074** | **0.185** | 0.092 | 0.208 | **0.070** | **0.178** | 0.083 | 0.194 | **0.073** | **0.185** | 0.087 | 0.195 | **0.069** | **0.173** |
| | 24 | 0.081 | 0.189 | **0.080** | **0.189** | 0.086 | 0.203 | **0.072** | **0.181** | 0.093 | 0.206 | **0.085** | **0.197** | 0.123 | 0.243 | **0.119** | **0.229** | 0.146 | 0.266 | **0.105** | **0.218** | 0.117 | 0.237 | **0.104** | **0.224** | 0.118 | 0.222 | **0.094** | **0.192** |
| | 48 | **0.118** | **0.230** | 0.127 | 0.238 | 0.130 | 0.248 | **0.110** | **0.220** | 0.160 | 0.273 | 0.279 | 0.360 | 0.225 | 0.333 | **0.189** | **0.301** | 0.275 | 0.372 | **0.177** | **0.285** | 0.207 | 0.321 | **0.174** | **0.287** | 0.186 | 0.272 | **0.119** | **0.220** |
| | 96 | 0.165 | 0.272 | 0.167 | 0.275 | 0.207 | 0.311 | **0.167** | **0.269** | 1.337 | 0.918 | **0.410** | **0.447** | 0.356 | 0.425 | **0.261** | **0.364** | 0.496 | 0.518 | **0.285** | **0.370** | 0.260 | 0.363 | **0.234** | **0.340** | 0.309 | 0.352 | **0.142** | **0.248** |
| | Avg | 0.107 | 0.214 | 0.109 | 0.216 | 0.122 | 0.235 | **0.102** | **0.209** | 0.414 | 0.392 | **0.209** | **0.293** | 0.196 | 0.341 | **0.161** | **0.270** | 0.262 | 0.321 | **0.159** | **0.263** | 0.167 | 0.279 | **0.146** | **0.259** | 0.175 | 0.260 | **0.106** | **0.208** |
| PEMS04 | 12 | **0.080** | **0.185** | 0.085 | 0.193 | 0.079 | 0.192 | **0.072** | **0.175** | 0.094 | 0.199 | **0.084** | **0.189** | 0.103 | 0.215 | **0.097** | **0.208** | 0.123 | 0.242 | **0.096** | **0.205** | 0.095 | 0.210 | **0.088** | **0.200** | 0.135 | 0.251 | **0.082** | **0.190** |
| | 24 | 0.102 | 0.213 | **0.093** | **0.199** | 0.095 | 0.212 | **0.085** | **0.193** | 0.130 | 0.236 | **0.102** | **0.211** | 0.157 | 0.273 | **0.144** | **0.258** | 0.196 | 0.309 | **0.139** | **0.252** | 0.126 | 0.248 | **0.111** | **0.230** | 0.180 | 0.290 | **0.095** | **0.208** |
| | 48 | 0.125 | 0.238 | **0.116** | **0.227** | 0.120 | 0.241 | **0.100** | **0.211** | 0.197 | 0.296 | **0.134** | **0.247** | 0.265 | 0.363 | **0.225** | **0.330** | 0.371 | 0.445 | **0.217** | **0.321** | 0.176 | 0.300 | **0.143** | **0.264** | 0.240 | 0.342 | **0.125** | **0.243** |
| | 96 | 0.141 | 0.254 | **0.136** | **0.247** | 0.157 | 0.279 | **0.136** | **0.250** | 0.328 | 0.398 | **0.178** | **0.288** | 0.406 | 0.462 | **0.298** | **0.390** | 0.647 | 0.591 | **0.294** | **0.384** | 0.529 | 0.567 | **0.184** | **0.301** | 0.335 | 0.418 | **0.166** | **0.288** |
| | Avg | 0.112 | 0.223 | **0.107** | **0.216** | 0.113 | 0.231 | **0.098** | **0.207** | 0.187 | 0.282 | **0.124** | **0.234** | 0.233 | 0.328 | **0.191** | **0.296** | 0.334 | 0.393 | **0.186** | **0.290** | 0.232 | 0.331 | **0.132** | **0.249** | 0.222 | 0.325 | **0.117** | **0.232** |
| PEMS07 | 12 | 0.067 | 0.163 | **0.062** | **0.163** | 0.064 | 0.169 | **0.055** | **0.150** | 0.067 | 0.167 | **0.060** | **0.158** | 0.077 | 0.176 | **0.069** | **0.172** | 0.097 | 0.204 | **0.076** | **0.188** | 0.105 | 0.222 | **0.068** | **0.169** |
| | 24 | 0.083 | 0.185 | **0.074** | **0.174** | 0.084 | 0.193 | **0.069** | **0.170** | 0.098 | 0.201 | **0.081** | **0.185** | 0.130 | 0.244 | **0.117** | **0.227** | 0.152 | 0.270 | **0.116** | **0.225** | 0.141 | 0.252 | **0.113** | **0.227** | 0.130 | 0.248 | **0.090** | **0.191** |
| | 48 | 0.103 | 0.204 | **0.102** | 0.207 | 0.108 | 0.212 | **0.080** | **0.181** | 1.511 | 1.012 | **0.205** | **0.309** | 0.241 | 0.326 | **0.204** | **0.306** | 0.303 | 0.385 | **0.201** | **0.297** | 0.179 | 0.294 | **0.116** | **0.227** |
| | 96 | 0.130 | 0.230 | 0.131 | 0.237 | 0.142 | 0.247 | **0.107** | **0.210** | 1.860 | 1.138 | **0.250** | **0.346** | 0.393 | 0.439 | **0.300** | **0.380** | 0.556 | 0.539 | **0.302** | **0.370** | 0.291 | 0.374 | **0.255** | **0.339** | 0.253 | 0.355 | **0.161** | **0.273** |
| | Avg | 0.096 | 0.195 | **0.092** | 0.195 | 0.100 | 0.205 | **0.078** | **0.178** | 0.884 | 0.629 | **0.149** | **0.249** | 0.211 | 0.302 | **0.174** | **0.272** | 0.275 | 0.351 | **0.173** | **0.267** | 0.196 | 0.290 | **0.155** | **0.259** | 0.167 | 0.280 | **0.109** | **0.215** |
| PEMS08 | 12 | 0.076 | 0.179 | **0.070** | **0.173** | 0.093 | 0.211 | **0.074** | **0.184** | 0.087 | 0.193 | **0.078** | **0.184** | 0.091 | 0.203 | **0.085** | **0.192** | 0.102 | 0.218 | **0.082** | **0.188** | 0.098 | 0.205 | **0.085** | **0.195** | 0.125 | 0.240 | **0.081** | **0.189** |
| | 24 | 0.107 | 0.211 | **0.098** | **0.207** | 0.134 | 0.256 | **0.101** | **0.217** | 0.132 | 0.242 | **0.110** | **0.220** | 0.145 | 0.258 | **0.127** | **0.240** | 0.168 | 0.279 | **0.122** | **0.231** | 0.145 | 0.260 | **0.116** | **0.231** | 0.161 | 0.277 | **0.098** | **0.210** |
| | 48 | 0.153 | 0.248 | **0.123** | **0.230** | 0.140 | 0.256 | **0.100** | **0.208** | 1.649 | 1.082 | **0.228** | **0.335** | 0.253 | 0.351 | **0.202** | **0.311** | 0.321 | 0.406 | **0.199** | **0.305** | 0.197 | 0.311 | **0.153** | **0.271** | 0.219 | 0.326 | **0.134** | **0.251** |
| | 96 | 0.136 | 0.236 | 0.137 | 0.237 | 0.210 | 0.318 | **0.156** | **0.261** | 1.571 | 0.986 | **0.302** | **0.397** | 0.402 | 0.466 | **0.277** | **0.385** | 0.614 | 0.581 | **0.287** | **0.379** | 0.439 | 0.493 | **0.216** | **0.331** | 0.335 | 0.416 | **0.199** | **0.318** |
| | Avg | 0.118 | 0.218 | **0.107** | **0.212** | 0.144 | 0.260 | **0.108** | **0.217** | 0.860 | 0.626 | **0.179** | **0.284** | 0.232 | 0.320 | **0.176** | **0.282** | 0.301 | 0.371 | **0.172** | **0.276** | 0.220 | 0.317 | **0.142** | **0.257** | 0.210 | 0.315 | **0.128** | **0.242** |
| Solar | 96 | 0.192 | 0.233 | **0.185** | **0.231** | 0.207 | 0.268 | **0.202** | **0.250** | 0.210 | 0.253 | **0.202** | **0.243** | 0.226 | 0.290 | **0.218** | **0.273** | 0.211 | 0.271 | **0.210** | **0.256** | 0.193 | 0.257 | 0.193 | 0.259 | 0.211 | **0.236** | 0.198 | 0.253 |
| | 192 | 0.223 | **0.253** | 0.215 | 0.255 | 0.241 | 0.293 | **0.240** | **0.276** | 0.240 | 0.277 | **0.225** | **0.268** | 0.244 | 0.277 | **0.294** | 0.235 | 0.272 | 0.237 | 0.281 | 0.224 | 0.287 | 0.256 | 0.281 | 0.221 | 0.274 |
| | 336 | 0.238 | 0.263 | **0.218** | **0.262** | 0.257 | 0.303 | **0.243** | **0.282** | 0.258 | 0.291 | **0.246** | **0.287** | 0.259 | 0.304 | **0.248** | **0.291** | 0.257 | 0.296 | **0.237** | **0.281** | 0.249 | 0.298 | 0.236 | 0.293 | 0.288 | 0.292 | **0.222** | **0.280** |
| | 720 | 0.234 | 0.264 | **0.210** | **0.263** | 0.251 | 0.297 | **0.281** | **0.281** | 0.255 | 0.291 | **0.238** | **0.285** | 0.249 | 0.295 | **0.245** | **0.291** | 0.246 | 0.293 | **0.233** | **0.278** | 0.522 | 0.539 | **0.235** | **0.294** | 0.285 | 0.295 | **0.218** | **0.271** |
| | Avg | 0.222 | **0.253** | 0.207 | 0.253 | 0.239 | 0.290 | **0.231** | **0.272** | 0.241 | 0.278 | **0.228** | **0.271** | 0.244 | 0.298 | **0.238** | **0.286** | 0.240 | 0.288 | **0.229** | **0.272** | 0.299 | 0.344 | **0.222** | **0.283** | 0.260 | 0.273 | **0.212** | **0.270** |

