# OpenReview forum: "Towards Accurate Time Series Forecasting via Implicit Decoding"
_NeurIPS.cc/2025/Conference — NeurIPS 2025 poster_

### Official Review · Reviewer_ZFna · 2025-06-29

**Clarity:** 3
**Significance:** 2
**Originality:** 3
**Rating:** 4
**Confidence:** 3

**Summary:**

This paper addresses a limitation in current time series forecasting (TSF) models: while existing methods mainly focus on effectively encoding historical data, they often overlook the decoding (forecasting) phase, which translates learned information into future predictions. To tackle this, the authors propose the Implicit Forecaster (IF) — a novel decoding module that improves the forecasting phase by predicting the future series as a combination of implicit frequency-based components. Instead of forecasting each future time point independently, IF models the time series globally by predicting the amplitude and phase of multiple constituent waves, leveraging Fourier analysis for signal reconstruction. Extensive experiments show that integrating IF into mainstream TSF models consistently improves their forecasting accuracy across diverse real-world datasets, setting new state-of-the-art results. Overall, the work highlights the importance of refining the forecasting stage in TSF pipelines and demonstrates the effectiveness of a frequency-domain approach to better capture global trends and periodic dynamics.

**Questions:**

1. I am quite curious about the scalability of this method. Could it be applied in some time series foundation models (e.g., \[1\]\[2\])?

[1] Timer: Generative Pre-trained Transformers Are Large Time Series Models

[2] Unified Training of Universal Time Series Forecasting Transformers

**Ethical Concerns:**

["NO or VERY MINOR ethics concerns only"]

**Final Justification:**

I think authors have addressed my concerns. So I raise my score to 4 to support this paper.

**Limitations:**

yes

**Quality:**

3

**Strengths And Weaknesses:**

Strengths:

1. The Implicit Forecaster is well motivated, combining frequency-domain analysis with implicit decomposition, and the design choices (e.g., amplitude and phase heads, skip-connections) are thoughtfully justified.

2. The authors conduct extensive experiments on 13 diverse, real-world datasets, demonstrating consistent performance gains over strong baselines, which shows the method’s generality and practical relevance.

3. The paper is well written, well structured, and easy to follow. The figures and tables effectively illustrate key points and results.

Weaknesses:

1. The choice of a fixed-size frequency pool might not be optimal across all domains. Some discussion on adaptive or learnable frequency selection could make the approach more flexible.

2. The authors should add more visualization results to demonstrate the effectiveness of the model design. For example, I suggest the authors to provide the visualization results for both Frequency prediction and Amplitude prediction.

---

> ### Author Rebuttal · Authors · 2025-07-31
>
> Dear Reviewer ZFna. We appreciate your positive comments and valuable suggestions regarding our paper. We respond to each of your points and questions as follows.
>
> ---
>
> > **W1. The choice of ... approach more flexible.**
>
> Thank you for your insightful and thoughtful comment. We agree that the use of a fixed-size frequency pool may not be optimal across all the data domains. In fact, we are adding discussions on the possibility of learnable frequency selection in the appendix of the revised paper, where we explore directions such as:
> - **Gated Frequency Pool**: One natural idea is that we can scan through the entire training dataset and find the most significant frequencies under a fixed input length based on the average amplitude (a similar method has been used in Koopa [1]). Then, combined with the amplitude of each frequency of the current input, it is possible to use a simple neural network and a gating mechanism to filter out less contributive frequencies in the output, thereby deactivating the corresponding weights in AHead and PHead by masking to achieve an adaptive frequency pool. Note that the computational complexity of the gating mechanism must be lower bounded by the cumulative cost of the pruned frequencies; otherwise, the adaptive frequency selection will fail to bring any real efficiency improvement.
>
> [1] Koopa: Learning Non-stationary Time Series Dynamics with Koopman Predictors. In The Thirty-seventh Annual Conference on Neural Information Processing Systems. 2023.
>
> > **W2. The authors should ... and Amplitude prediction.**
>
> We have added more visualization results in the appendix of our revised paper to demonstrate both phase and amplitude predictions, including the results showing the effect of errors in phase prediction on the final time-domain output. However, we regret that we are unable to share the results at this time as NeurIPS guidelines explicitly prohibit any figures during the rebuttal period. Nevertheless, we assure the reviewers that these visualizations and the corresponding analyses will be included in the final version of our paper.
>
> > **Q1. I am quite ... models (e.g., [1][2])?**
>
> We have reviewed the time series foundation models mentioned in the references. From the perspective of model architecture, our method can indeed be applied to such models. Since these models typically learn a unified time series encoder representation, which is well-suited to be used as inputs to our IF decoder for prediction. We are currently running experiments to verify this, but due to time limitations, we have not obtained the final results. Once the experiments are completed, the performance of IF on time series foundation models will be reported in the final version of our paper as "further generality analysis".
>
> ---
>
> We hope our clarifications addressed your concerns, and we are looking forward to answering any further questions.

---

> > ### Comment · Reviewer_ZFna · 2025-08-03
> >
> > Thank you very much for your response, which has addressed most of my concerns. I have raised my score to 4 to support this paper.

---

> > > ### Author Response · Authors · 2025-08-05
> > >
> > > Dear Reviewer ZFna,
> > >
> > > We really appreciate your recognition of our work and your kind words, and we are happy to hear that your concerns have been addressed. Again, thank you for your valuable suggestions which have undoubtedly contributed to improving the quality of our paper.
> > >
> > > Many thanks,
> > >
> > > The authors of #28364

---

### Official Review · Reviewer_rnsD · 2025-07-01

**Clarity:** 2
**Significance:** 2
**Originality:** 3
**Rating:** 4
**Confidence:** 5

**Summary:**

The paper introduces a model-agnostic framework for time series forecasting that identifies prediction failure cases through uncertainty estimation and corrects them from local and global perspectives. The method aims to enhance predictive performance while maintaining extensibility, with experimental validation across multiple scenarios.

**Questions:**

- The paper does not use the widely adopted traffic dataset in this
field. Is this omission due to the dataset's large number of variables? Could the authors further analyze how the number of variables impacts the computational overhead of the frequency- domain method (e.g., the relationship between variable count and computation time/resource consumption)?
- Some models (e.g., TimesNet) exhibit poor performance on certain datasets. Could the authors investigate the root causes of these performance differences? Are they driven by dataset characteristics, model architectural design, or other factors?
- Further justification is needed to clarify how the proposed method differs from and improves upon prior scale-based decomposition approaches (e.g., TimesNet, TimeMixer). Could the authors supplement experiments on TimeMixer to validate the method's competitiveness?

**Ethical Concerns:**

["NO or VERY MINOR ethics concerns only"]

**Final Justification:**

According to the authors’ rebuttal and additional experiments, I increase my score. Thanks for the additional explanation. Please revise your paper in the final version.

**Limitations:**

yes.

**Paper Formatting Concerns:**

no major formatting issues

**Quality:**

3

**Strengths And Weaknesses:**

Strengths and Weaknesses: Strengths:
- As a flexible decoding model, the method can adapt to various time
series forecasting models, exhibiting model-agnostic
   characteristics;
- Experimental results demonstrate effectiveness in forecasting tasks.
Weaknesses:
- The frequency-domain approach introduces additional computational overhead, which may be more pronounced in multi-variable datasets, but the paper lacks analysis on how variable counts affect computational costs;
- The proposed new perspective essentially aligns with the conventional decomposition-based approach (e.g., decomposing time series into periodic components), a methodology already prevalent in works like TimesNet and TimeMixer[1]. The frequency-based MLP method (FIST[2]) is also not novel. The paper provides insufficient comparative analysis or case studies to justify its novelty;
- No comparison with the strong baseline TimeMixer, which also employs decomposition-based methods.
[1] Timemixer: Decomposable multiscale mixing for time series forecasting
[2] FITS: Modeling Time Series with Parameters

---

> ### Author Rebuttal · Authors · 2025-07-31
>
> Dear Reviewer rnsD. Thank you for the time and effort you dedicated to reviewing our paper! Below, we provide detailed clarifications on each of your points.
>
> ---
>
> > **W1. The frequency-domain approach ... affect computational costs;**
>
> Thank you for raising this concern regarding computational complexity. Below, we show that our frequency-domain approach (IF) does not incur additional computational overhead compared to conventional point-wise decoders in multivariate settings.
>
> Let $ \hat{Y} \in \mathbb{R}^{L \times N} $ denote the predicted future time series over horizon $L$ with $N$ variates. We use the following standard asymptotic notation:
>
> - $\mathcal{O}(g)$: upper bound up to constant factor, i.e., $f(n) \in \mathcal{O}(g(n))$ means there exist constants $c, n_0$ such that $f(n) \leq c \cdot g(n)$ for all $n \geq n_0$.
>
> Then:
> 1. **Time Complexity for Point-wise Decoders:**
>
>     Conventional point-wise decoders predict values individually for each variate at each time point, with different time points corresponding to different output neurons, and sharing the same neurons across variates. Thus, their computational complexity is directly proportional to both the forecasting length $L$ and the number of variates $N$, which is simply bounded by $\mathcal{O}(N \times L)$, as they must produce $N \times L$ independent predictions.
>
> 2. **Time Complexity for our IF Decoder:**
>
>     In contrast, our IF decoder predicts frequency-domain features, with a fixed frequency pool size $P \geq L$ and $N$ variates. As discussed in Paragraph **Computational efficiency**, IF uses the symmetric properties of frequency-domain signals, resulting in $\frac{3}{2}P + 3$ output neurons per variate, independent of the forecasting length $L$, **and shared across variates**. Therefore, the forward pass complexity of IF is simply bounded by $\mathcal{O}(N \times P)$. However, after predicting frequency-domain features, an inverse DFT (IDFT) is applied to reconstruct the time-domain signal. The IDFT per variate can be implemented efficiently with complexity $\mathcal{O}(P\log P)$ using FFT-based algorithms. Hence, for all $N$ variates, the complexity of IDFT is $\mathcal{O}(N \times P\log P)$. Combining both terms, the total time complexity of IF = decoder + IDFT complexity = $\mathcal{O}(N \times P + N \times P\log P)$ = $\mathcal{O}(N \times P\log P)$.
>
> Comparing the time complexity of point-wise decoders and our IF decoder, we observe that both decoders scale linearly with the number of variates. Thus, our approach does not introduce additional computational overhead related to variate counts.
>
> > **W2. The proposed new ... justify its novelty;**
>
> We appreciate your comments concerning the perspective of our method. However, we would like to clarify that the core novelty of our method does **not** lie in adopting a decomposition-based forecasting approach. As we stated in Lines 39-46 of Section 1 **Introduction**, our method is *motivated* by decomposition forecasting. Instead, our key contribution is to **improve the decoding stage of time series forecasting** and inspire further work on this point, which is often overlooked in previous methods. Specifically, the proposed IF decoder is designed to enhance the **globality and pattern clarity** of the predicted time series.
>
> Compared with prior decomposition-based approaches such as TimesNet and TimeMixer, our method offers several advantages:
> - **Global Prediction**: Since frequency features are computed for the entire time series, our IF decoder can **have better global views of the prediction**, which allows it to express long-range temporal patterns in a more compact and coherent form. In contrast, TimeMixer and TimesNet still make predictions in a point-wise manner.
> - **Precise Pattern Control**: We think previous decomposition-based forecasting methods can hardly guarantee that their predictions align accurately with trend-seasonal components. However, in our approach, since each output neuron in IF is specialized in predicting one particular wave at a certain frequency, it inherently **ensures that the predicted components have well-defined, distinct periodic patterns** (i.e., low frequencies correspond to trends, and high frequencies represent seasonalities), offering component distinctness.
> - **Efficiency**: Given that our method predicts underlying dynamics as frequency components, each component can be **compactly** represented by its amplitude and phase, thereby enabling more efficient forecasting. In contrast, previous decomposition-based approaches still rely on point-wise predictions for each component in the time domain, which leads to higher computational overhead.
>
> Thanks for the references to prior frequency-based methods such as FITS. However, the core innovation with FITS is in learning the **mappings** of input-output time series in the complex plane using complex-valued linear layers, while our approach is focused on predicting encoded time series representations into **frequency features** of various dynamics to construct the future series (such dynamics can be longer than the future series), which is a critical distinction that contributes to the novelty of our approach. Furthermore, our method is compatible with a broad range of backbones and can be plugged into existing forecasting models.
>
> > **W3. No comparison with ... employs decomposition-based methods.**
>
> For each model type, we select the **most competitive method** as the baseline. MLP-based models like TimeMixer have fallen behind SOFTS [1], which is why we did not choose TimeMixer as our baseline. Moreover, there is a compatibility issue between IF and TimeMixer. Given that **TimeMixer separately learns and predicts decomposed long-short term patterns with different predictors** (i.e., it has a list of encoder representations and a list of predicted patterns), we can hardly find a fused encoder representation (i.e., a unified `enc_out`) that is suitable to be fed into our IF decoder. We regret not being able to provide the corresponding experimental results.
>
> [1] SOFTS: Efficient Multivariate Time Series Forecasting with Series-Core Fusion. In The Thirty-eighth Annual Conference on Neural Information Processing Systems. 2024.
>
> > **Q1P1. The paper does ... number of variables?**
>
> Both the Traffic dataset and the PEMS datasets are widely used time series forecasting datasets for transportation. However, the PEMS datasets have four different subsets (e.g., PEMS03, PEMS04, PEMS07, and PEMS08), which **cover a broader range of scenarios**, including varying road types, traffic densities, and sensor distributions. Moreover, the PEMS datasets exhibit **super long-term patterns mixed with rapid, short-term changes**, making them more suitable for evaluating a decoder’s ability to express ultra-long-range fluctuations and represent compounded temporal dynamics. Therefore, we chose the PEMS datasets to better demonstrate our model's strengths in handling more challenging forecasting scenarios.
>
> > **Q1P2. Could the authors ... computation time/resource consumption)?**
>
> We have analysed the relationship between the variate count and computation time/resource consumption. Please kindly refer to the response: **W1.**.
>
> > **Q2. Some models (e.g., ... or other factors?**
>
> Thank you for your valuable question. We would like to clarify that, for experimental fairness, we did not perform any hyperparameter tuning for the baseline models or the models augmented with IF throughout the entire experiments. Instead, to ensure consistency and accurately assess our approach, all models used the **same hyperparameters** before and after applying IF, which are provided by their original code repositories. This constraint could affect the performance of IF, making it sub-optimal. For example, TimesNet’s relatively poor performance on certain datasets (e.g., ETTh1 and ETTh2). We believe this is largely due to its hyperparameter configurations, where the model dimension and the feedforward dimension are set to 32, which is very small and will limit IF’s learning capacity.
>
> > **Q3P1. Further justification is ... (e.g., TimesNet, TimeMixer).**
>
> Please kindly refer to the response: **W2.**.
>
> > **Q3P2. Could the authors ... the method's competitiveness?**
>
> Please kindly refer to the response: **W3.**.
>
> ---
>
> We have clarified and discussed the above points further in the paper revisions and hope we have addressed your comments. If you have any further questions and comments, we'd be delighted to address them.

---

> > ### Comment · Area_Chair_8VBH · 2025-08-05
> > **Engaging with the rebuttal**
> >
> > Dear reviewer,
> >
> > The discussion phase is soon coming to and end. It will be great if you could go over the rebuttal and discuss with the authors if you still have outstanding concerns. Thank you for being part of the review process.
> >
> > Regards,
> >
> > Area Chair

---

> ### Author Response · Authors · 2025-08-05
> **We are working on more settings and will update you once the results are available.**
>
> Dear Reviewer rnsD,
>
> Thank you for your valuable feedback and the time dedicated to reviewing our work. We are also working on more settings and will update you once the results are available, for reasons below.
>
> During our rebuttal, we mentioned that there is a compatibility issue between IF and TimeMixer. Given that TimeMixer separately learns and predicts decomposed long-short term patterns with different predictors (i.e., it has a list of encoder representations and a list of predicted patterns), we can hardly find a fused encoder representation (i.e., a unified ```enc_out```) that is suitable to be fed into our IF decoder. Nonetheless, **we recently found that with proper modification, our method can be applied to the competitive TimeMixer to further boost its performance.**
>
> We will update you once the results are available - thank you so much!
>
> Many thanks,
>
> The authors of #28364

---

> ### Author Response · Authors · 2025-08-08
> **More results on TimeMixer and Traffic dataset**
>
> Dear Reviewer rnsD,
>
> We sincerely appreciate your patience in awaiting our follow-up and additional results, and thank you again for your valuable reviews and for kindly pointing out our oversights!
>
> > **No comparison with the strong baseline TimeMixer, which also employs decomposition-based methods.**
>
> We appreciate your concerns regarding the absence of a comparison with the strong baseline TimeMixer. In our original rebuttal, we noted a compatibility issue between IF and TimeMixer due to TimeMixer's separate handling of long short-scale patterns. However, **we have since addressed this limitation and conducted experiments on TimeMixer to further validate our method's competitiveness**.
>
> Specifically, we revised the integration strategy by **carefully replacing each individual predictor in TimeMixer's Future-Multipredictor-Mixing (FMM) block with our IF decoder**. In TimeMixer, each predictor is responsible for forecasting long or short-scale patterns in a point-wise manner. By substituting these predictors with IF, we enabled a more global and accurate prediction at each temporal scale while preserving the multiscale design of TimeMixer (note that we reduced the size of IF accordingly to minimize the computational overhead). The corresponding experimental results are as follows:
>
> | TimeMixer (MSE) | Original  | w/ IF     |
> | --------------- | --------- | --------- |
> | ETTh1           | 0.530     | **0.494** |
> | ETTh2           | 0.194     | **0.187** |
> | ETTm1           | **0.466** | 0.473     |
> | ETTm2           | 0.146     | **0.142** |
> | ECL             | **0.179** | 0.182     |
> | Weather         | 0.224     | **0.224** |
> | Exchange        | 0.410     | **0.373** |
> | ILI             | 3.234     | **2.917** |
> | PEMS03          | 0.180     | **0.126** |
> | PEMS04          | 0.204     | **0.128** |
> | PEMS07          | 0.173     | **0.129** |
> | PEMS08          | 0.217     | **0.129** |
> | Solar           | 0.211     | **0.201** |
> | Promotion       |           | **12.8%** |
>
> The above table shows that **integrating IF into TimeMixer consistently improves forecasting performance on most datasets**, particularly in ETTh1 and PEMS datasets. While the gains are modest on certain datasets, the overall performance demonstrates that IF can serve as a strong enhancement to decomposition-based architectures like TimeMixer.
>
> > **The paper does not use the widely adopted traffic dataset in this field.**
>
> We would like to provide the results on the Traffic dataset for **one baseline from each model type (i.e., MLP, Transformer, RNN, and CNN)**, as shown below:
>
> | Traffic (MSE) | Original | w/ IF     |
> | ------------- | -------- | --------- |
> | SOFTS         | 0.433    | **0.429** |
> | Crossformer   | 0.575    | **0.503** |
> | SegRNN        | 0.639    | **0.573** |
> | TimesNet      | 0.614    | **0.593** |
> | Promotion     |          | **6.8%**  |
>
> From the above results, it is evident that the **proposed IF decoder can boost the performance of baselines across different model types on the widely adopted Traffic dataset**, further indicating its generality (due to the time constraint, we could not include all baselines in this round of experiments, yet we will present the complete results in the final version of our revised paper).
>
> > **Could the authors further analyze how the number of variables impacts the computational overhead of the frequency- domain method (e.g., the relationship between variable count and computation time/resource consumption)?**
>
> In our original rebuttal, we theoretically showed that IF scales linearly with the number of variables and thus does not incur additional computational overhead on this dimension compared to point-wise decoders. **We now empirically verify this by comparing IF with a linear decoder under varying variable counts**, using the PEMS07 dataset, which has the largest number of variables (#883) among all our datasets. We gradually increased the number of variables used for testing, and the results are as follows:
>
> | Var. Count | IF Speed (ms/iter) | Linear Speed (ms/iter) |
> | - | - | - |
> | 100        | 20.30| 17.57 |
> | 200        | 20.40  | 17.38|
> | 300        | 20.99  | 18.33|
> | 400        | 21.99 | 18.65 |
> | 500        | 23.35  | 18.27|
> | 600        | 25.14              | 18.15|
> | 700        | 25.70              | 18.03|
> | 800        | 27.07              | 18.51 |
>
> **The above table confirms that IF scales linearly with the variable count (approx. 1.04 ms per 100 variables, R²=0.96)**, while the point-wise linear decoder exhibits a slow trend (approx. 0.11ms per 100 variables). These results corroborate our claim that IF does not incur **superlinear** overhead w.r.t. the number of variables, consistent with our theoretical analysis.
>
> ***
> Please let us know if we have resolved your concerns – thank you!
>
> Many thanks,
>
> Authors of #28364

---

### Official Review · Reviewer_P8Px · 2025-07-02

**Clarity:** 4
**Significance:** 3
**Originality:** 4
**Rating:** 5
**Confidence:** 5

**Summary:**

This paper proposes a novel decoding module, Implicit Forecaster (IF), which decomposes time series into frequency components (amplitude, phase, frequency) for implicit prediction and reconstructs future series via inverse DFT. Experiments show IF consistently boosts mainstream models (e.g., 23.9% MSE reduction on PEMS) and achieves SOTA on 13 real-world datasets, especially for long-term and mixed-frequency patterns.

**Questions:**

Seen in the weak part

**Ethical Concerns:**

["NO or VERY MINOR ethics concerns only"]

**Limitations:**

Seen in the weak part

**Quality:**

4

**Strengths And Weaknesses:**

strong
(1) It introduces a novel innovative decoding: Implicit Forecaster (IF) replaces point-wise prediction with frequency-based implicit modeling, addressing the lack of global views. And it could seamlessly integrates with Transformers, RNNs, CNNs, etc., with significant improvements (e.g., 16.2% avg. MSE drop on Crossformer).
(2) By setting P≥L, IF directly models ultra-low-frequency waves, enhancing long-term forecasting. It also gives the interpretability analysis to reveal key frequencies learned by the model.
(3) It covers 13 datasets and 7 baselines, validating generality (e.g., 10.9% MAE improvement on SegRNN). And rigorous ablations (decoder replacements, skip-connection, window length) isolate each design choice.
(4) The O(P) time and less memory than a Transformer decoder while improving accuracy.

weak:
(1) Lacks theoretical comparison between frequency vs. point-wise prediction.
(2) Theoretical analysis is shallow, the convergence or approximation error of implicit reconstruction is not explored.
(3) There is no adaptive frequency selection because it needs to predict all frequencies, which may be inefficient. And manual P setting may not match optimal data specific frequencies. While sin/cos decomposition stabilises phase prediction, it fails to model abrupt phase shifts (e.g., traffic congestion spikes in PEMS), causing residual errors in high frequency components (Figure 4).
(4) Channel-independent forecasting overlooks inter-variable frequency correlations (e.g., synchronized power grid fluctuations), potentially losing multivariate dynamics.

---

> ### Author Rebuttal · Authors · 2025-07-30
>
> Dear Reviewer P8Px. We sincerely thank you for recognizing our work and for giving valuable, thorough reviews. We provide detailed responses to each of your points as follows:
>
> ---
>
> > **W1. Lacks theoretical comparison ... vs. point-wise prediction.**
>
> Many thanks for the insightful feedback on the theoretical aspects of our work. We have provided theoretical results demonstrating that, compared to conventional point-wise decoders, the proposed Implicit Forecaster offers better decoder conditioning and stability under noise. Please kindly refer to **Theorem 1** in response: **Reviewer Cqck, W1**, for details.
>
> > **W2, W3P1. Theoretical analysis is ... may be inefficient.**
>
> We appreciate your concerns regarding these valuable points. We agree that our current framework lacks mechanisms for adaptive frequency selection, which may limit the forecasting efficiency. Nevertheless, it is possible to directly discard less contributive frequencies (e.g., ultra-high-frequency waves) from IF's frequency pool for improving efficiency. To support this, we have established a theoretical foundation demonstrating that our IF decoder offers precise control over approximation and prediction errors from frequency truncation. Below, we quantify reconstruction errors of IF:
>
> **Theorem 2** (Approximation Error from Frequency Truncation)**.** *Let $Y\_{:,n} \in \mathbb{R}^L$ be a real-valued time series and let $z^{(n)} = \mathcal{F}(Y\_{:,n}) \in \mathbb{C}^L$ be its Discrete Fourier Transform (DFT). Let $S \subseteq \\{0, 1, \dots, L-1\\}$ denote the frequency indices retained by the model, and define the truncated spectrum:*
> $$
> \hat{z}\_k^{(n)} =
> \begin{cases}
> z\_k^{(n)} & \text{if } k \in S, \\\\
> 0 & \text{otherwise}.
> \end{cases}
> $$
> *Let $\hat{Y}\_{:,n} = \mathcal{F}^{-1}(\hat{z}^{(n)})$ be the reconstructed signal from the truncated spectrum. Then the squared reconstruction error satisfies:*
> $$
> \\| Y\_{:,n} - \hat{Y}\_{:,n} \\|\^2 = \sum\_{k \notin S} |z\_k^{(n)}|^2.
> $$
>
> This result quantifies the exact loss incurred when we truncate the frequency pool in IF. The reconstruction error is just the sum of the squared magnitudes of the discarded frequency coefficients. IF provides transparent and controllable approximation: you can directly trade off forecasting efficiency and accuracy by selecting which frequencies to keep.
>
> *Proof.* Let $e := Y\_{:,n} - \hat{Y}\_{:,n}$ denote the reconstruction error in the time domain. Since the inverse DFT is a linear transformation, and $\mathcal{F}^{-1}$ is unitary up to a scaling constant, we can analyze the error in the frequency domain.
>
> Define the error spectrum:
> $$
> e = \mathcal{F}^{-1}(z^{(n)} - \hat{z}^{(n)}),
> \quad \text{so} \quad
> \\|e\\|\_2^2 = \left\\| \mathcal{F}^{-1}(z^{(n)} - \hat{z}^{(n)}) \right\\|\_2^2.
> $$
> We use the unitary DFT, i.e., both $\mathcal{F}$ and $\mathcal{F}^{-1}$ include $1/\sqrt{L}$, now apply **Parseval's identity** for the DFT:
> $$
> \\| \mathcal{F}^{-1}(w) \\|\_2^2 =  \\| w \\|\_2^2
> \quad \text{for any } w \in \mathbb{C}^L.
> $$
> Applying this identity to the difference vector $z^{(n)} - \hat{z}^{(n)}$, we obtain:
> $$
> \\|e\\|\_2^2 = \left\\| \mathcal{F}^{-1}(z^{(n)} - \hat{z}^{(n)}) \right\\|\_2^2
> = \\| z^{(n)} - \hat{z}^{(n)} \\|\_2^2.
> $$
> By the definition of $\hat{z}^{(n)}$, we have:
> $$
> z\_k^{(n)} - \hat{z}\_k^{(n)} =
> \begin{cases}
> 0 & \text{if } k \in S, \\\\
> z\_k^{(n)} & \text{if } k \notin S,
> \end{cases}
> $$
> so:
> $$
> \\| z^{(n)} - \hat{z}^{(n)} \\|\_2^2 = \sum\_{k \notin S} |z\_k^{(n)}|^2.
> $$
> Putting it all together:
> $$
> \\| Y\_{:,n} - \hat{Y}\_{:,n} \\|\_2^2 = \sum\_{k \notin S} |z\_k^{(n)}|^2.
> $$
> This completes the proof.
>
> > **W3P2. And manual P ... data specific frequencies.**
>
> Indeed, a manual $P$ can not precisely align with the optimal data specific frequencies, as stated in Section 4.3 **Limitation Analysis**. However, a larger $P$ can provide our IF decoder with a **finer frequency granularity**, which makes the predictable frequencies have the chance to align closer with the frequencies that best fit the dataset, thereby alleviating the **Spectral Leakage** issue (We have explicitly demonstrated and discussed this issue in the appendix of the revised paper.).
>
> > **W3P3. While sin/cos decomposition ... components (Figure 4).**
>
> Thank you for raising this insightful observation. To address your concern, we theoretically show that our IF decoder is stable under amplitude and phase estimation/prediction errors.
>
> **Theorem 3** (Stability to Estimation Error)**.** *Let the implicit forecaster predict $\hat{z}\_k^{(n)} = \hat{A}\_k^{(n)} e^{j \hat{\phi}\_k^{(n)}}$ for $k = 1, \dots, P$, and let the ground truth coefficients be $z\_k^{(n)} = A\_k^{(n)} e^{j \phi\_k^{(n)}}$. Define:*
> $$
> \delta\_A^{(n)} := \\| \hat{\mathbf{A}}^{(n)} - \mathbf{A}^{(n)} \\|\_2,
> \quad
> \delta\_{\phi}^{(n)} := \\| \hat{\boldsymbol{\phi}}^{(n)} - \boldsymbol{\phi}^{(n)} \\|\_2,
> \quad
> A\_{\max}^{(n)} := \max\_k |A\_k^{(n)}|.
> $$
> *Then, the total error satisfies:*
> $$
> \left\\| \mathcal{F}^{-1}(\hat{\mathbf{z}}^{(n)}) - \mathcal{F}^{-1}(\mathbf{z}^{(n)}) \right\\|\_2
> \leq \delta\_A^{(n)} + A\_{\max}^{(n)} \cdot \delta\_{\phi}^{(n)}.
> $$
> This theorem shows how small errors in amplitude and phase estimates translate into total reconstruction error. The bound is linear in the errors, and decouples the contributions of amplitude and phase. This proves that IF degrades gracefully with bounded amplitude/phase prediction errors, with stability improving as $P$ increases.
>
> *Proof.* We decompose the per-frequency error:
> $$
> \hat{z}\_k^{(n)} - z\_k^{(n)} = \underbrace{(\hat{A}\_k^{(n)} - A\_k^{(n)}) e^{j \hat{\phi}\_k^{(n)}}}\_{\text{amplitude error}} + \underbrace{A\_k^{(n)} \left(e^{j \hat{\phi}\_k^{(n)}} - e^{j \phi\_k^{(n)}}\right)}\_{\text{phase error}}.
> $$
> Applying the triangle inequality for complex magnitudes:
> $$
> \left| \hat{z}\_k^{(n)} - z\_k^{(n)} \right| \leq \left| \hat{A}\_k^{(n)} - A\_k^{(n)} \right| \underbrace{\left| e^{j \hat{\phi}\_k^{(n)}} \right|}\_{=1} + \left| A\_k^{(n)} \right| \left| e^{j \hat{\phi}\_k^{(n)}} - e^{j \phi\_k^{(n)}} \right|.
> $$
> Using the sharp inequality $ |e^{j\alpha} - e^{j\beta}| \leq |\alpha - \beta| $ (which holds because the chordal distance on the unit circle is bounded by angular distance):
> $$
> \left| \hat{z}\_k^{(n)} - z\_k^{(n)} \right| \leq \left| \hat{A}\_k^{(n)} - A\_k^{(n)} \right| + \left| A\_k^{(n)} \right| \left| \hat{\phi}\_k^{(n)} - \phi\_k^{(n)} \right|.
> $$
> Define $\mathbf{e}^{(n)} = \hat{\mathbf{z}}^{(n)} - \mathbf{z}^{(n)}$. The vector norm satisfies:
> $$
> \\| \mathbf{e}^{(n)} \\|\_2 \leq \left\\| (\hat{\mathbf{A}}^{(n)} - \mathbf{A}^{(n)}) \odot e^{j\hat{\boldsymbol{\phi}}^{(n)}} \right\\|\_2 + \left\\| \mathbf{A}^{(n)} \odot (e^{j\hat{\boldsymbol{\phi}}^{(n)}} - e^{j\boldsymbol{\phi}^{(n)}}) \right\\|\_2.
> $$
> For the first term:
> $$
> \left\\| (\hat{\mathbf{A}}^{(n)} - \mathbf{A}^{(n)}) \odot e^{j\hat{\boldsymbol{\phi}}^{(n)}} \right\\|\_2 = \| \hat{\mathbf{A}}^{(n)} - \mathbf{A}^{(n)} \|_2 = \delta_A^{(n)}
> $$
> since $|e^{j\hat{\phi}_k^{(n)}}| = 1$ for all $k$.
>
> For the second term:
> $$
> \left\\| \mathbf{A}^{(n)} \odot (e^{j\hat{\boldsymbol{\phi}}^{(n)}} - e^{j\boldsymbol{\phi}^{(n)}}) \right\\|\_2 \leq \max\_k |A\_k^{(n)}| \cdot \\| e^{j\hat{\boldsymbol{\phi}}^{(n)}} - e^{j\boldsymbol{\phi}^{(n)}} \\|\_2 \leq A\_{\max}^{(n)} \\| \hat{\boldsymbol{\phi}}^{(n)} - \boldsymbol{\phi}^{(n)} \\|_2
> $$
> where the final inequality uses $ |e^{j\alpha} - e^{j\beta}| \leq |\alpha - \beta| $ component-wise. Thus:
> $$
> \left\\| \mathbf{A}^{(n)} \odot (e^{j\hat{\boldsymbol{\phi}}^{(n)}} - e^{j\boldsymbol{\phi}^{(n)}}) \right\\|\_2 \leq A\_{\max}^{(n)} \delta\_{\phi}^{(n)}.
> $$
> Combining both terms:
> $$
> \\| \mathbf{e}^{(n)} \\|\_2 \leq \delta\_A^{(n)} + A\_{\max}^{(n)} \delta\_{\phi}^{(n)}.
> $$
> Since the inverse DFT is unitary ($\ell\_2$-norm preserving):
> $$
> \left\\| \mathcal{F}^{-1}(\hat{\mathbf{z}}^{(n)}) - \mathcal{F}^{-1}(\mathbf{z}^{(n)}) \right\\|\_2 = \\| \mathbf{e}^{(n)} \\|\_2 \leq \delta\_A^{(n)} + A\_{\max}^{(n)} \delta\_{\phi}^{(n)}.
> $$
> This completes the proof.
>
> > **W4. Channel-independent forecasting overlooks ... losing multivariate dynamics.**
>
> This is an excellent suggestion! In our method, we improve the forecasting performance by enhancing the prediction's globality. However, when forecasting, it is also worthwhile to consider **the correlation between the variates of the output series**. This is undoubtedly a valuable research direction to pursue further.
>
> ---
>
> We have refined our paper by including the above theoretical results and wish our responses have addressed your comments. Once again, thank you for carefully reading our paper and kindly pointing out our oversights! We'd be happy to respond to further questions.

---

> > ### Comment · Area_Chair_8VBH · 2025-08-05
> > **Engaging with the rebuttal**
> >
> > Dear reviewer,
> >
> > The discussion phase is soon coming to and end. It will be great if you could go over the rebuttal and discuss with the authors if you still have outstanding concerns. Thank you for being part of the review process.
> >
> > Regards,
> >
> > Area Chair

---

### Official Review · Reviewer_Cqck · 2025-07-03

**Clarity:** 3
**Significance:** 3
**Originality:** 3
**Rating:** 4
**Confidence:** 3

**Summary:**

This paper studies the problem of time series forecasting, and proposes an implicit forecaster to predict decomposed constituent waves. The proposed method addresses the overlooked forecasting phase, and the empirical performance is demonstrated on various real-world datasets.

**Questions:**

Is there any other intuition why the proposed decoding strategy is better beyond DFT properties?

Maybe I missed it, how is the frequency pool determined? Is the performance sensitive to the its size?

Minor: It would be better to repeat the experiments with different runs to exam the robustness of the results and report "error bars".

**Ethical Concerns:**

["NO or VERY MINOR ethics concerns only"]

**Final Justification:**

Most of my concerns are addressed, I therefore increased my rating to 4.

**Limitations:**

yes

**Paper Formatting Concerns:**

Table 1 is too crowded and hard to read.

**Quality:**

3

**Strengths And Weaknesses:**

The paper is generally well-written and easy to follow. The idea is intuitive and focuses on the decoding phase of time series forecasting, which is less explored than the encoder.

While the method is intuitive and demonstrated empirically, the paper lack of theoretical justification, which limits the applicability of this method. More discussion would enhance the significance of this work, and guide the application more broadly.

The proposed implicit forecaster is plug-and-play, and leads to an inherent dependence and potential sensitivity to the quality of the encoder. It would be better to have a thorough investigation on the robustness with mediocre encoders.

---

> ### Author Rebuttal · Authors · 2025-07-30
>
> Dear Reviewer Cqck. Thank you for your insightful and valuable comments. We would like to address each of your concerns in detail below.
>
> ---
>
> > **W1. While the method ... application more broadly.**
>
> We appreciate your feedback regarding the theoretical justification of our work. Below, we discuss a theoretical advantage of the proposed **Implicit Forecaster (IF)** in the long-term forecasting setting:
>
> Let $X \in \mathbb{R}^{T \times N}$ denote the historical input time series with $N$ variates and length $T$, and $Y \in \mathbb{R}^{L \times N}$ denote the ground-truth future series over horizon $L$. We denote the DFT of each variate $n \in \\{1, \dots, N\\}$ by $z^{(n)} = \mathcal{F}(Y\_{:,n}) \in \mathbb{C}^{L}$. IF reconstructs predictions as:
> $$
> \hat{Y}\_{:,n} = \mathcal{F}^{-1}(\hat{A}^{(n)} \cdot e^{j\hat{\phi}^{(n)}})\_{:L},\quad\forall n \in \\{1, \dots, N\\},
> $$
> where $\hat{A}^{(n)} \in \mathbb{R}\_{\geq 0}^{P}$ and $\hat{\phi}^{(n)} \in [-\pi, \pi]^P$ are the predicted amplitudes and phases for variate $n$, with frequency pool size $P \geq L$. We formalize IF’s improved stability and conditioning compared to point-wise decoders.
>
> **Theorem 1** (Decoder Conditioning Advantage)**.** *Let $Y\_{:,n} \in \mathbb{R}^L$ be the ground-truth future series for variate $n$, and let $h \in \mathbb{R}^D$ be a shared latent representation. Suppose:*
>
> - *A point-wise decoder (pw) predicts $\hat{Y}\_{\mathrm{pw},:,n} = \Phi^{(n)}h$ with decoder matrix $\Phi^{(n)} \in \mathbb{R}^{L \times D}$;*
> - *An implicit decoder (IF) predicts $\hat{Y}\_{\mathrm{IF},:,n} = G \hat{z}^{(n)}$ using a **unitary** inverse DFT matrix $G \in \mathbb{C}^{L \times L}$ (i.e.,* $G^\*G = I\_L$*) and learned complex coefficients $\hat{z}^{(n)} \in \mathbb{C}^L$;*
> - *The latent representation is perturbed by noise:* $h = h^* + \eta$*, where $\eta \sim \mathcal{N}(0,\sigma^2 I\_D)$;*
>
> *Assume the decoders are unbiased:* $\Phi^{(n)} h^* = Y\_{:,n}$ *and $G z^{(n)} = Y\_{:,n}$. Then, the expected prediction error satisfies:*
> $$
> \mathbb{E} \left[ \\| \hat{Y}\_{\mathrm{pw},:,n} - Y\_{:,n} \\|^2 \right]
> \geq \sigma\_{\min}^2(\Phi^{(n)}) \cdot \mathbb{E} \left[ \\| \hat{Y}\_{\mathrm{IF},:,n} - Y\_{:,n} \\|^2 \right],
> $$
> *where $\sigma\_{\min}(\Phi^{(n)})$ is the smallest singular value of $\Phi^{(n)}$.*
>
> The theorem shows that the error of the point-wise decoder is lower bounded by a factor of our IF decoder’s error, where the factor is the smallest singular value of the point-wise decoder’s weight matrix. If this value is small (i.e., the decoder is ill-conditioned), the bound becomes loose, so the theorem does not guarantee that IF always performs better. However, the key advantage of IF lies in its prediction stability: since it uses a unitary inverse DFT matrix, the decoder does not amplify latent noise. In contrast, point-wise decoders can suffer significant error amplification if not well-conditioned. Thus, while **Theorem 1** alone doesn’t prove IF is always superior, it highlights the robustness of IF in noisy or ill-conditioned settings.
>
> *Proof.* We analyze the variance of prediction error under noise. For the point-wise decoder:
> $$
> \hat{Y}\_{\mathrm{pw},:,n} = \Phi^{(n)} (h^* + \eta) = Y\_{:,n} + \Phi^{(n)} \eta,
> $$
> so the expected squared error is:
> $$
> \mathbb{E} \left[ \\| \hat{Y}\_{\mathrm{pw},:,n} - Y\_{:,n} \\|^2 \right]
> = \mathbb{E} \left[ \\| \Phi^{(n)} \eta \\|^2 \right]
> = \sigma^2 \cdot \mathrm{Tr} \left( \Phi^{(n)} (\Phi^{(n)})^\top \right).
> $$
> For the IF decoder, let $\hat{z}^{(n)} = z^{(n)} + \tilde{\eta}$ where $\tilde{\eta} \in \mathbb{C}^L$ has zero-mean independent components with $\mathbb{E}[\tilde{\eta}\tilde{\eta}^\*] = \sigma^2 I\_L$. Then:
> $$
> \hat{Y}\_{\mathrm{IF},:,n} = G (\hat{z}^{(n)}) = G (z^{(n)} + \tilde{\eta}) = Y\_{:,n} + G \tilde{\eta},
> $$
> and since $G$ is unitary:
> $$
> \mathbb{E} \left[ \\| \hat{Y}\_{\mathrm{IF},:,n} - Y\_{:,n} \\|^2 \right]
> = \mathbb{E} \left[ \\| G \tilde{\eta} \\|^2 \right]
> = \mathbb{E} \left[ \tilde{\eta}^* G^* G \tilde{\eta} \right]
> = \mathbb{E} \left[ \tilde{\eta}^* \tilde{\eta} \right]
> = \sigma^2 L.
> $$
> Now for the point-wise decoder, we have:
> $$
> \mathrm{Tr} \left( \Phi^{(n)} (\Phi^{(n)})^\top \right)
> = \sum\_{i=1}^L \sigma\_i^2 \geq L \cdot \sigma\_{\min}^2(\Phi^{(n)}),
> $$
> where $\sigma\_i$ are the singular values of $\Phi^{(n)}$. This implies:
> $$
> \mathbb{E} \left[ \\| \hat{Y}\_{\mathrm{pw},:,n} - Y\_{:,n} \\|^2 \right]
> \geq \sigma^2 L \cdot \sigma\_{\min}^2(\Phi^{(n)})
> = \sigma\_{\min}^2(\Phi^{(n)}) \cdot \mathbb{E} \left[ \\| \hat{Y}\_{\mathrm{IF},:,n} - Y\_{:,n} \\|^2 \right].
> $$
> This completes the proof.
>
> We have provided more theoretical analysis of our work. Please kindly refer to the response: **Reviewer P8Px, W2, W3P1 and W3P3**.
>
> > **W2. The proposed implicit ... with mediocre encoders.**
>
> Thank you for this valuable suggestion. As part of our robustness study, we have applied the proposed IF module to a previous time series encoder, **Informer** [1]. This encoder can be reasonably considered as **mediocre** compared to recent SOTA encoders. The corresponding results are presented below:
>
> | Informer (MSE) | Original | w/ IF |
> | - | - | - |
> | ETTh1 | 1.165 | **0.697** |
> | ETTh2 | 0.321 | **0.290** |
> | ETTm1 | 0.717 | **0.618** |
> | ETTm2 | 0.242 | **0.191** |
> | ECL | 0.353 | **0.262** |
> | Weather | **0.589** | 0.620 |
> | Exchange | **1.523** | 1.544 |
> | ILI | 6.075 | **5.106** |
> | PEMS03 | 0.162 | **0.122** |
> | PEMS04 | 0.121 | **0.106** |
> | PEMS07 | 0.185 | **0.126** |
> | PEMS08 | 0.150 | **0.115** |
> | Solar | 0.224 | **0.206** |
> | Promotion | | **16.9%** |
>
> The above results demonstrate that our IF module can consistently improve Informer's performance across different datasets, confirming its robustness and general applicability in mediocre time series encoders.
>
> [1] Informer: Beyond Efficient Transformer for Long Sequence Time-Series Forecasting. Proceedings of the AAAI Conference on Artificial Intelligence. 2021.
>
> > **Q1. Is there any ... beyond DFT properties?**
>
> Indeed, there is another intuitive explanation that supports the effectiveness of the proposed decoding strategy. Our IF decoder employs a **pattern-separated** prediction strategy, enabling the model to focus on simpler, more predictable underlying dynamics rather than the complex, intertwined original series. In our approach, the key point is that these dynamics are represented as frequency components, such that **each output neuron is specialized in predicting one particular wave at a certain frequency**. Unlike previous point-wise decomposition forecasting methods, which cannot reliably guarantee that their predictions align accurately with trend-seasonal components, IF has **component distinctness**. This is because our frequency-based prediction strategy inherently ensures that the predicted components have well-defined, distinct periodic patterns (i.e., low frequencies correspond to trends, and high frequencies represent seasonalities), offering better performance and interpretability.
>
> > **Q2P1. Maybe I missed ... frequency pool determined?**
>
> We apologize for not clearly describing how the frequency pool size is determined, as we set it as a **hyperparameter**. As discussed in Paragraph **Frequency Pool**, the pool size $P$ must be greater than or equal to the forecasting length $L$ to ensure signal reconstruction completeness. Therefore, in our experiments, we set the pool size to a default value of $P=720$ for almost all cases, as this size sufficiently covers the length of major fluctuations observed across all datasets and is greater than most forecasting lengths (Note that the maximum forecasting length is 720.). We have clarified this point in our revised version of the paper.
>
> > **Q2P2. Is the performance ... the its size?**
>
> The forecasting performance is sensitive to IF's pool size. As explained in Paragraph **Frequency Pool**, we believe that a larger pool size can benefit the model from finer frequency granularity. This allows the predicted frequencies to align closer with the frequencies that best fit the dataset, thereby alleviating the **Spectral Leakage** issue (We have clearly demonstrated and discussed this point in the appendix of the revised paper.). Moreover, a larger pool size enables IF to have access to ultra-low-frequency components, enhancing predictions of long-term patterns. Below, we present the MSE results of shrinking the pool size from its default $P=720$ to the prediction length $L$:
>
> | Pool Size (MSE) | P | L |
> | - | - | - |
> | ETTh2 | **0.188** | 0.191 |
> | Exchange | **0.286** | 0.339 |
> | ILI | **2.387** | 2.941 |
> | PEMS-08 | **0.103** | 0.145 |
> | Performance | **100%**  | 83.7% |
>
> The table shows that reducing the pool size leads to a noticeable decline in performance, indicating that a broader frequency pool can enhance IF’s ability to access lower frequency components and represent diverse waveforms, facilitating better performance.
>
> > **Q3. Minor: It would ... report "error bars".**
>
> Thank you for pointing this out. In fact, we have already tested the robustness of our model across five runs with different random seeds and reported the corresponding error bars in the appendix of our paper (see **Supplementary Material**). We kindly direct your attention to this supplementary information.
>
> ---
>
> We have revised our paper by adding the above theoretical results and experimental analysis. We hope our responses and clarifications addressed your concerns, and we are happy to deal with any further questions.

---

> > ### Comment · Area_Chair_8VBH · 2025-08-05
> > **Engaging with the rebuttal**
> >
> > Dear reviewer,
> >
> > The discussion phase is soon coming to and end. It will be great if you could go over the rebuttal and discuss with the authors if you still have outstanding concerns. Thank you for being part of the review process.
> >
> > Regards,
> >
> > Area Chair

---

> > ### Comment · Reviewer_Cqck · 2025-08-06
> >
> > Thanks for the detailed reply, and I appreciate the theoretical insights and robustness check of experiments. Most of my concerns are addressed, I therefore increased my rating to 4.

---

> > > ### Author Response · Authors · 2025-08-06
> > >
> > > Dear Reviewer Cqck,
> > >
> > > We really appreciate your recognition of our work and your kind words, and we are happy to hear that your concerns have been addressed. Again, thank you for your valuable suggestions which have undoubtedly contributed to improving the quality of our paper.
> > >
> > > Many thanks,
> > >
> > > The authors of #28364

---

> ### Author Response · Authors · 2025-08-05
> **We follow your suggestion to repeat the experiments with different runs to report "error bars" to validate the significance of performance gain.**
>
> Dear Reviewer Cqck,
>
> Thank you again for your thoughtful comments.
>
> > It would be better to repeat the experiments with different runs to exam the robustness of the results and report "error bars".
>
> To verify whether the improvement is statistically significant, **we have also follow your suggestion to repeat the experiments with different runs to report "error bars"**. In addition, we performed a **paired t-test to further validate the significance of performance gain** of our method (due to the extensive repeated experiments, we have to present these results at this time).
>
> ---
>
> Specifically, we applied our IF module to the Transformer backbone by modifying only the output layer (i.e., `nn.Linear` $\rightarrow$ `ImplicitForecaster`), and kept all hyperparameters identical (including the random seed) before and after applying IF to ensure a strictly fair comparison. For the paired t-test, we test the hypothesis that Transformer with IF achieves significantly lower MSE than without IF. The experimental results averaged over 4 prediction lengths are as follows:
>
> | Transformer (MSE) | Original              | w/ IF                 | p-value               | Significance |
> | ----------------- | --------------------- | --------------------- | --------------------- | ------------ |
> | ETTh1             | 0.539 $\pm$ 0.001     | **0.504 $\pm$ 0.002** | $1.98 \times 10^{-5}$ | ***          |
> | ETTh2             | 0.200 $\pm$ 0.001     | **0.188 $\pm$ 0.001** | $4.68 \times 10^{-4}$ | ***          |
> | ETTm1             | 0.508 $\pm$ 0.001     | **0.493 $\pm$ 0.003** | $1.04 \times 10^{-3}$ | **           |
> | ETTm2             | 0.155 $\pm$ 0.001     | **0.148 $\pm$ 0.001** | $1.23 \times 10^{-5}$ | ***          |
> | ECL               | 0.162 $\pm$ 0.001     | **0.159 $\pm$ 0.000** | $5.71 \times 10^{-3}$ | **           |
> | Weather           | 0.228 $\pm$ 0.000     | **0.220 $\pm$ 0.000** | $2.15 \times 10^{-5}$ | ***          |
> | Exchange          | 0.406 $\pm$ 0.004     | **0.297 $\pm$ 0.003** | $1.31 \times 10^{-5}$ | ***          |
> | ILI               | 2.985 $\pm$ 0.013     | **2.770 $\pm$ 0.053** | $1.03 \times 10^{-2}$ | *            |
> | PEMS03            | 0.123 $\pm$ 0.001     | **0.095 $\pm$ 0.000** | $9.71 \times 10^{-6}$ | ***          |
> | PEMS04            | 0.152 $\pm$ 0.022     | **0.099 $\pm$ 0.006** | $3.67 \times 10^{-2}$ | *            |
> | PEMS07            | 0.103 $\pm$ 0.001     | **0.081 $\pm$ 0.001** | $6.70 \times 10^{-6}$ | ***          |
> | PEMS08            | 0.176 $\pm$ 0.011     | **0.104 $\pm$ 0.002** | $1.67 \times 10^{-3}$ | **           |
> | Solar             | **0.188 $\pm$ 0.001** | 0.189 $\pm$ 0.001     | $5.44 \times 10^{-1}$ | n.s.         |
>
> (Significance levels: * $p < 0.05$, ** $p < 0.01$, *** $p < 0.001$, n.s. **not significant**)
>
> From the above table, it is evident that the performance of our method is considerably robust, and **the proposed IF module is stably and statistically significantly outperforms the point-wise prediction method (in 12 out of 13 datasets)**.
>
> Many thanks,
>
> Authors of #28364

---

### Decision · Program_Chairs · 2025-09-17

**Decision:**

Accept (poster)

**Comment:**

This work proposes a decoding module named Implicit Forecaster. It decomposes time series into frequency components (amplitude, phase, frequency) for implicit prediction and reconstructs future series via inverse DFT.  Instead of forecasting each future time point independently, the forecaster models the time series globally by predicting the amplitude and phase of multiple constituent waves, leveraging Fourier analysis for signal reconstruction.

The paper received 4 reviews and the reviewers appreciated the overall quality of the work and the experimental results. Despite this there were several major points of contention that the reviewers pointed out:

* A lack of theoretical analysis.
* The choice of a fixed-size frequency pool might not be optimal across all domains. Also this might get inefficient for larger domains
* The work closely resembles conventional decomposition-based approaches and the differences between them and the proposed method was not very clear.

The rebuttal provided was detailed and the authors did a great job especially in providing the critical missing theoretical links. The new experiments on the Traffic data set and the comparison with TimeMixer was also well appreciated. Overall, all the reviewers agreed on accepting the paper and I concur. I request the authors to include the new results and the theoretical analysis from the rebuttal in the final version of the paper.